# System Identification and Nonlinear Model Predictive Control with Collision Avoidance Applied in Hexacopters UAVs

**DOI:** 10.3390/s22134712

**Published:** 2022-06-22

**Authors:** Luis F. Recalde, Bryan S. Guevara, Christian P. Carvajal, Victor H. Andaluz, José Varela-Aldás, Daniel C. Gandolfo

**Affiliations:** 1SISAu Research Group, Facultad de Ingeniería y Tecnologías de la Información y Comunicación, Universidad Tecnológica Indoamérica, Ambato 180103, Ecuador; fernandorecalde@uti.edu.ec; 2Instituto de Automática, Universidad Nacional de San Juan-CONICET, Av. San Martín Oeste 1109, San Juan J5400ARL, Argentina; bguevara@inaut.unsj.edu.ar (B.S.G.); cpcarvajal@inaut.unsj.edu.ar (C.P.C.); dgandolfo@inaut.unsj.edu.ar (D.C.G.); 3Departamento de Eléctrica y Electrónica, Universidad de las Fuerzas Armadas–ESPE, Sangolquí 171103, Ecuador; vhandaluz1@espe.edu.ec; 4Department of Electronic Engineering and Communications, University of Zaragoza, 44003 Teruel, Spain

**Keywords:** system identification, model predictive control, obstacles avoidance, hexacopter UAV, system constraints, optimization

## Abstract

Accurate trajectory tracking is a critical property of unmanned aerial vehicles (UAVs) due to system nonlinearities, under-actuated properties and constraints. Specifically, the use of unmanned rotorcrafts with accuracy trajectory tracking controllers in dynamic environments has the potential to improve the fields of environment monitoring, safety, search and rescue, border surveillance, geology and mining, agriculture industry, and traffic control. Monitoring operations in dynamic environments produce significant complications with respect to accuracy and obstacles in the surrounding environment and, in many cases, it is difficult to perform even with state-of-the-art controllers. This work presents a nonlinear model predictive control (NMPC) with collision avoidance for hexacopters’ trajectory tracking in dynamic environments, as well as shows a comparative study between the accuracies of the Euler–Lagrange formulation and the dynamic mode decomposition (DMD) models in order to find the precise representation of the system dynamics. The proposed controller includes limits on the maneuverability velocities, system dynamics, obstacles and the tracking error in the optimization control problem (OCP). In order to show the good performance of this control proposal, computational simulations and real experiments were carried out using a six rotary-wind unmanned aerial vehicle (hexacopter—DJI MATRICE 600). The experimental results prove the good performance of the predictive scheme and its ability to regenerate the optimal control policy. Simulation results expand the proposed controller in simulating highly dynamic environments that showing the scalability of the controller.

## 1. Introduction

### 1.1. Motivation

In recent years, the field of robotics has experienced exponential growth due to fast technological advances. The applications are not restricted to the industrial field, as nowadays robots are equipped with sophisticated and intelligent algorithms that are useful for performing complex task in unstructured and natural environments with a high degree of autonomy. The integration between robots and humans through service robotics has the objective of facilitating activities of daily life or improving repetitive tasks. As such, we can find mobile manipulators that are used to load transportation in structured environments [1,2], robotic systems that help people during their rehabilitation process [3] and aerial robotic vehicles that are useful in rescue or transportation operations [4]. Robot applications can be divided into four main groups: *(i) Learning robots,* which are robotic systems for whom the main objective is to improve learning and stimulation processes [5,6]. *(ii) Construction robots,* which are systems that can be programmed to develop complex repetitive tasks, where the main contributions are the decrease in production time and the prevention of accidents in dangerous workplaces [7,8]. *(iii) Field robots,* which show a variety of improvements in mining industries, agricultural and livestock, increasing the economy of the population [9,10]. *(iv) Military robots,* which are designed to perform exploration, security, rescue, among other applications in hostile environments that can be dangerous for humans, and these applications are highly developed using unmanned aerial vehicles (UAVs) [11].

The use of unmanned aerial vehicles have grown in a wide range of fields and has been extensively studied by the scientific community due to the versatility and the possibility to perform the fully autonomous tasks that humans are unable to. The extremely agility of the UAVs with four or more rotors has attracted attention due to their ability to vertically take off and land in confined or difficult to access spaces, which eases construction and enables having a higher payload capacity, which are useful in critical missions such as transportation, rescue, search, surveillance and disaster assistance. In these applications, the execution time is an extremely important feature due to the task only being executed for high-speed trajectories. To take advantage of UAVs, it is necessary to guarantee the safe execution of the application; however, dynamic environments are spaces with highly probabilities of collisions, where obstacle information is unknown at the beginning of the task, and thus the UAV must be capable of avoiding the obstacles using online-scheme information. Obstacle avoidance is an important feature that guarantees the correct functionality of UAV applications. In this context, research efforts are still needed to safely introduce UAVs as a technological tool with great potential to solve various current problems such as a collision avoidance control structure that can perform high-speed trajectories in dynamic environments.

### 1.2. Related Work

Depending on the environment, UAV applications can be divided in two groups: *(i) Outdoor environments* where many studies have employed UAVs with onboard sensors, e.g., Mellinger et al. presented collaborative grasping to logistic applications [12]; Bircher et al. performed path planning for structural inspection using UAVs [13]; search and rescue operations were developed by Oetter- shagen et al. in [14]; and an application consisting of tasks related to physical interaction with the environment was developed by Garimella et al. [15]. *(ii) Indoor environments* are the counterpart applications, developed in cluttered indoor experiments, examples of which include the study by Song and Hsu who presented navigation based on factor graph optimization (FGO) [16]; Paredes et al., who developed a hybrid acoustic and optical positioning system for accurate 3D movements [17]; and Sandino et al., who proposed an autonomous navigation system using partial observable and uncertainty sensor measurements [18]. Due to the different types of UAVs, these systems have the ability to perform a wide range of applications that are emerging. The most popular configuration of multi-rotors is the quadrotor UAV, which is applied in a variety of aerial fields, e.g., Radoglou–Grammatiski et al., who presented the compilation of the most important applications for precision agriculture using quadrotors [19]; and Gupta et al., who presented the advances of UAVs and the applications in transportation systems [20]. This configuration only presents four motors attached to the mechanical frame generating a limiting power that restricts the fields of use, especially for rescue and transportation tasks. However, nowadays the design of a multi-rotor with more than four rotors is a rapidly growing industry because multi-rotor configuration has presented important advances due to its high maneuverability and transportation capacity [21,22]. More works were conducted by Belmonte et al., who designed an octocopter for the inspection of mobile cranes [23]; Phong-Nguyen et al., who presented a sliding mode control algorithm with a fuzzy inference system considering variable gains for hexacopters [24]; and Ali et al., who developed adaptive backstepping for the attitude and altitude of coaxial octorotors [25].

The use of more rotors can produce high costs in production, a considerably increased size of the UAV and complex dynamics. Nowadays, it still a challenge to design effective controllers that ensure good performance in multi-rotors at high-speed trajectories. Precise trajectory tracking is an important feature for multi-rotors operating in real-world environments due to the considerable size of the structure and possible collision with obstacles. The main reasons that make this configuration more complex than the others are presented below: *(i) System actuation:* An inherently under-actuated system due to there only being four control inputs over the six degrees of freedom (DOF). *(ii) Dynamics behavior:* high nonlinearities produced by translational and rotational dynamics due to the considerable size of the system. *(iii) Dynamic model mismatch:* Unmodeled dynamics, uncertainties and disturbances produced for high-speed trajectories and the possibility of external forces such as wind in outdoor environments. All of these problems incited the special attention of researchers from different fields. As a consequence, a wide variety of control strategies have been proposed for the problem of trajectory tracking for hexacopters. Initially, some linear control methods were presented, and these works considered small angles’ assumptions that are necessary for linear control techniques, and the most common structures were presented by Alaimo et al., who developed a proportional–integral and derivative controller (PID) under linear assumptions in UAVs with an hexacopter configuration [26] and Salim et al. presented optimum linear control that stabilizes the attitude of a micro-hexacopter in indoor environments [27].

Linear models are only valid around the hover position, providing slow movements where the controllers cannot guarantee the convergence to the desired trajectories. In order to improve the performance and to satisfy the requirements of high-speed trajectories, a large number of studies have used the principles of nonlinear control, e.g., Chen et al., who proposed a nonlinear trajectory controller for UAVs based on backstepping and nonlinear observer [28] and Wang et al. presented the sliding mode control (SMC) with a variable structure to generate the control inputs [29]. Recently, many authors have proposed using geometric tracking controllers, e.g., Lee et al. showed the results of trajectory tracking using a nonlinear controller developed in a Euclidean group SE(3) [30]; Mellinger et al. revealed the differential flatness properties in UAVs and the ability to derive the attitude, acceleration and angular rate [31]. The property of differential flatness improves the accurate trajectory tracking under high-speed references and it was demonstrated by [32,33]. In recent years, intelligent control approaches were used to developed advance control systems, and these schemes included artificial neural networks and fuzzy logic controllers that are highly used in multi-rotor UAVs; the results present interesting behaviors under disturbances and parameter changes [34,35,36,37].

Unlike multi-rotor control techniques, obstacle avoidance has not been highly developed by the research community. Obstacle avoidance is a relatively new field and most works have used the concept of artificial potential fields, a method which is based on the generation of repulsive and attractive forces that allow the movements of the system around obstacles. The literature presents the extensive applications with UAVs, but these works revealed the problem of a local minimal that cannot guarantee the execution of the task [38,39]. Nowadays, other methods have been proposed to solve the problem of obstacle avoidance, the most popular among which include: *(i) rapidly exploring random tree (RRT)*, a formulation which generates a tree rooted at the initial configuration of the system and using random samples to create edges between feasible points. The work by Achtelik et al., who presented an RRT algorithm with a modification rapidly exploring random belief tree (RRBT), which was used to generate the collision free path using quadrotor UAVs [40]; *(ii) The belief roadmap*, which is a probabilistic version of the roadmap, which uses the Kalman filter for the planning motion, and has the ability to generate trajectories by avoiding obstacles; this structure was presented by [41]. *(iii) The minimum snap trajectory generation* this algorithm includes navigation through corridors whilst considering constraints on velocities and accelerations, the solution to which is based on the generation of optimal trajectories that minimizes a functional cost confirmed by square norm of the snap, an application of which was presented in [31]. All the obstacle avoidance algorithms presented beforehand only consider the prior knowledge of the obstacle in the environment, whilst also requiring high computational times and complex operations.

Taking into account the problems of no-dynamics constraints and only the prior knowledge of environments, the results thus produce unfeasible control actions and problems in dynamic environments with highly probabilities of collisions during the task execution. To fully exploit hexacopters’ capabilities and take advantage of the available computation power, optimization-based control techniques are becoming suitable for real-time problems. Model predictive control (MPC) has been used in many fields of robotics due to the accuracy of its tracking trajectories, robust performance and the ability to include constraints in the controller scheme. In function of the field, the general applications can be divided into the following groups:Ground mobile robots: where Sani et al. presented nonlinear model predictive control (NMPC) to solve the competitive games between ground robots [42], another work undertaken developed by Subramanian et al. increased the ability to avoid dynamic obstacles [43], and finally, a leader–follower structure control using NMPC through visual information to mobile robots was presented by Ribeiro et al. [44].Robotic arms and manipulation: In the last year, this field of research has been the focus of many studies due to the complexity of the systems and the scalability of the NMPC structure. The work of Osman et al. presented a task-space controller based on (NMPC) to control a mobile manipulator 10 (DOF) [45].Aerial mobile robots: this field has also been the subject of a large number of studies due to the high use of aerial vehicles in real-world applications, some applications of which were studies Neunert et al., who developed an unconstrained nonlinear predictive model for generation and tracking trajectories for the AscTec Firefly hexacopter [46]; Aoki et al., who presented an NMPC for position and attitude control applied in a hexacopter without three of the six motor configurations [47]; and finally, the contribution of Tzoumanikas et al., a letter which presented the NMPC designed for micro aerial vehicles (MAVs) equipped with a robotic arm [48].

This kind of controller has had a huge impact on the field of robotics thanks to the predictive behavior and ability to introduce system constraints in the controller scheme. The main idea of this controller is to generate the control actions trying to minimize a cost function over a prediction horizon. The cost function is solved using constrained optimization techniques; however, the resolution of constrained optimization problems is computationally expensive. Due to the recent developments in hardware and nonlinear optimization solvers, nonlinear predictive control is computationally applicable in many fields and has received the attention of a hexacopter control field [49,50,51,52,53].

### 1.3. Main Contributions of This Work

Run NMPC structures are realizable in modern computers and guarantee the execution of complex tasks but still require more computational time than the schemes mentioned beforehand. It is necessary to show performance under tracking accuracy and the ability to extend this technique to dynamic environments in order to add obstacles in the optimization problem. With all the aforementioned controllers and the significant results in many areas, this work presents the nonlinear model predictive control to fast trajectory tracking in dynamic environments applied in hexacopter platforms. The controller best generates the control policy that moves the hexacopter to the reference trajectory while avoiding obstacles through the environment. To construct the approach, this work developed the important contributions presented below:System dynamics: a large number of research platforms cannot be controlled through torque commands; as such, this work developed a reduced dynamic model of the hexacopter using general control velocities, which is possible with the incorporation of low-level PID schemes in mathematical representation. The PIDs guarantee the required velocities generated by the high-level controllers. The mathematical representation was developed through an Euler–Lagrange formulation and the data-driven technique known as dynamic mode decomposition (DMDc). Both approaches were identified using experimental data from the (DJI MATRICE 600 PRO); furthermore, this work presents accuracy comparative results between both techniques and selects the best formulation to build the controller scheme.NMPC formulation: this includes the differential kinematics and the reduced dynamic representation; furthermore, due to the ability of the NMPC controller, the system constraints can be included in the optimization problem. The constraints included in this work are: control action limits, control rate of change, and the system dynamics and static and dynamic obstacles. The nonlinear optimization problem was solved using CasADI framework and the problem was transformed into a nonlinear programming problem (NLP) form using the direct multiple-shooting method. The CasADI optimization toolkit guarantees a fast convergence with the possibility of extending the solution to onboard hardware implementation.Results: the experiments were conducted in real-world outdoor environments, where the system has a model mismatch and external disturbance product by air flows and delay in system communications. The experiments were developed through the hexacopter platform (DJI MATRICE 600 PRO) where multiple reference trajectories were selected in order to verify the performance of the controller. The metrics used in the experiments are the following: tracking accuracy, computational time and disturbance rejection. Additionally, the simulation results show the scalability in simulated high dynamic environments, considering the identified dynamics and obstacles; these results will be used as a start point in future research due to the fact that the hexacopter aerial platform does not have any sensor to measure obstacles during the experiments.

### 1.4. Outline

This work is structured as follows. Section 2 presents the instantaneous kinematics with the reduced dynamic model and system identification and validation. Furthermore, the controller formulation is presented in this section, which consists of the nonlinear model predictive controller with an obstacles avoidance scheme. Section 3 shows the real-world experiments and simulation results; furthermore, considering the experiment results, this section exhibits a review of the important aspects and the future research direction. The conclusions of the work are presented in Section 4.

## 2. Materials and Methods

### 2.1. Hexacopter System Preliminaries

This section presents the instantaneous kinematics and the dynamic model. The dynamic model was formulated in maneuverability velocities space using two different formulations: the Euler–Lagrange and dynamic mode decomposition. After the formulation process, this work presents the validation results in order to verify the performance of each formulation.

Figure 1 shows the hexacopter platform DJI MATRICE 600 PRO, where the world-fixed inertial frame is represented by <I> with the following unit vectors $\left\{I_{x},I_{y},I_{z}\right\}$ and the body-fixed frame attached to hexacopter movements is defined by <B> with the unit vectors $\left\{B_{x},B_{y},B_{z}\right\}$, where the center of mass (CoM) is aligned; furthermore, the hexacopter is configured with six motors in the mechanical frame which allows the movement of the system.

#### 2.1.1. Kinematics Model

The robot has the ability to move through the inertial frame <I> and it is enabled to rotate only in yaw ψ defined in the vertical axis $B_{z}$, whilst the others’ angular rotations roll and pitch (ϕ,θ) are not considered in this work due to this hexacopter platform having a low level flight controller which guarantees the hover position. The position and orientation of the point to be controlled is defined by the vector $\eta ={\left[\begin{array}{cccc} \eta_{x} & \eta_{y} & \eta_{z} & \eta_{\psi } \end{array}\right]}^{T}\in R^{4}$ with respect to the frame <I>, and the following equation defines these elements in greater detail:(1)$$\left\{\begin{array}{c} \eta_{x}=\eta_{x_{0}}+d_{x}\cos\left(\psi \right)-d_{y}\sin\left(\psi \right)\\ \eta_{y}=\eta_{y_{0}}+d_{x}\sin\left(\psi \right)+d_{y}\cos\left(\psi \right)\\ \eta_{z}=\eta_{z_{0}}+d_{z}\\ \eta_{\psi }=\psi \end{array}\right.$$
where the values $\left(\eta_{x_{0}},\eta_{y_{0}},\eta_{z_{0}}\right)$ are the locations of the CoM and $\left(d_{x},d_{y},d_{z}\right)$ are the distances in the body-fixed frame <B> to the point of interest η. In order to know the evolution of the interest point, it is necessary to introduce the concept of the instantaneous kinematic, defining the time derivative as $\dot{\eta }=\frac{\partial \eta }{\partial t}\mu$. With these considerations, the velocity of the point is defined by the vector $\dot{\eta }={\left[\begin{array}{cccc} {\dot{\eta }}_{x} & {\dot{\eta }}_{y} & {\dot{\eta }}_{z} & {\dot{\eta }}_{\psi } \end{array}\right]}^{T}\in R^{4}$ with respect to the frame <I>; furthermore, the time derivative produces maneuverability velocities μ attached to the frame <B>. Due to the low-level PIDs controllers, the maneuverability vector is defined by $\mu ={\left[\begin{array}{cccc} \mu_{l} & \mu_{m} & \mu_{n} & \omega \end{array}\right]}^{T}\in R^{4}$, with the longitudinal velocities $\left(\mu_{l},\mu_{m},\mu_{n}\right)$ through the axis $\left(B_{x},B_{y},B_{z}\right)$ and the angular rate of change ω over unit vector $B_{z}$. Using the definition of instantaneous kinematics, the evolution can be defined by the following equation:(2)$$\left\{\begin{array}{c} {\dot{\eta }}_{x}=\mu_{l}\cos\left(\psi \right)-\mu_{m}\sin\left(\psi \right)-\left(d_{x}\sin\left(\psi \right)+d_{y}\cos\left(\psi \right)\right)\omega \\ {\dot{\eta }}_{y}=\mu_{l}\sin\left(\psi \right)+\mu_{m}\cos\left(\psi \right)+\left(d_{x}\cos\left(\psi \right)-d_{y}\sin\left(\psi \right)\right)\omega \\ {\dot{\eta }}_{z}=\mu_{n}\\ {\dot{\eta }}_{\psi }=\omega \end{array}\right.$$
the equation presented before can be written in a matrix form, defined as:(3)$$\left[\begin{array}{c} {\dot{\eta }}_{x}\\ {\dot{\eta }}_{y}\\ {\dot{\eta }}_{z}\\ {\dot{\eta }}_{\psi } \end{array}\right]=\left[\begin{array}{cccc} \cos(\psi ) & -\sin(\psi ) & 0 & -(d_{x}\sin\left(\psi \right)+d_{y}\cos\left(\psi \right))\\ \sin(\psi ) & \cos(\psi ) & 0 & \left(d_{x}\cos\left(\psi \right)-d_{y}\sin\left(\psi \right)\right)\\ 0 & 0 & 1 & 0\\ 0 & 0 & 0 & 1 \end{array}\right]\left[\begin{array}{c} \mu_{l}\\ \mu_{m}\\ \mu_{n}\\ \omega \end{array}\right]$$
where the expressions $-(d_{x}\sin\left(\psi \right)+d_{y}\cos\left(\psi \right))$ and $\left(d_{x}\cos\left(\psi \right)-d_{y}\sin\left(\psi \right)\right)$ represent the additional behavior considering the displacement of the point of interest. To simplify the notation, Equation (Equation 3) is expressed in the compact form in (Equation 4)
(4)$$\dot{\eta }\left(t\right)=J\left(\psi \left(t\right)\right)\mu \left(t\right)$$
where $J\left(\psi \left(t\right)\right)\in R^{4\times 4}$ is the Jacobian matrix which allows the linear mapping between the control maneuverability velocities μ to the evolution of the point of interest $\dot{\eta }$, Equation (Equation 3) is expressed in vector-function form as:(5)$$\dot{\eta }\left(t\right)=f_{k}\left(\psi \left(t\right),\mu \left(t\right)\right)$$

Considering that $\ddot{\eta }\left(t\right)=\frac{d}{dt}\dot{\eta }$, the following relation was developed:(6)$$\begin{array}{c} \left[\begin{array}{c} {\ddot{\eta }}_{x}\\ {\ddot{\eta }}_{y}\\ {\ddot{\eta }}_{z}\\ {\ddot{\eta }}_{\psi } \end{array}\right]=\left[\begin{array}{cccc} \cos(\psi ) & -\sin(\psi ) & 0 & -(d_{x}\sin\left(\psi \right)+d_{y}\cos\left(\psi \right))\\ \sin(\psi ) & \cos(\psi ) & 0 & \left(d_{x}\cos\left(\psi \right)-d_{y}\sin\left(\psi \right)\right)\\ 0 & 0 & 1 & 0\\ 0 & 0 & 0 & 1 \end{array}\right]\left[\begin{array}{c} \dot{\mu_{l}}\\ \dot{\mu_{m}}\\ \dot{\mu_{n}}\\ \dot{\omega } \end{array}\right]+\cdots \\ \left[\begin{array}{cccc} -\omega \sin(\psi ) & -\omega \cos(\psi ) & 0 & -\omega (d_{x}\cos\left(\psi \right)-d_{y}\sin\left(\psi \right))\\ \omega \cos(\psi ) & -\omega \sin(\psi ) & 0 & \omega (d_{x}\sin\left(\psi \right)+d_{y}\cos\left(\psi \right))\\ 0 & 0 & 0 & 0\\ 0 & 0 & 0 & 0 \end{array}\right]\left[\begin{array}{c} \mu_{l}\\ \mu_{m}\\ \mu_{n}\\ \omega \end{array}\right] \end{array}$$
it can be written in compact form as $\ddot{\eta }\left(t\right)=J\left(\psi \right)\dot{\mu }\left(t\right)+\dot{J}\left(\psi ,\omega \right)\mu \left(t\right)$, where $\dot{\mu }={\left[\begin{array}{cccc} \dot{\mu_{l}} & \dot{\mu_{m}} & \dot{\mu_{n}} & \dot{\omega } \end{array}\right]}^{T}\in R^{4}$ is the vector of maneuverability accelerations, and this formulation will be used in the dynamic model section.

#### 2.1.2. Dynamic Model

This section presents the dynamic model, which has been developed considering low-level PIDs controllers that only generate longitudinal movements in $\left\{I_{x},I_{y},I_{z}\right\}$ and the angular rotation through $B_{z}$. The dynamic model is useful for guaranteeing the stability of the proposed controller in real-world applications.

##### Euler–Lagrange Formulation

One of the approaches to find the mathematical model is the Euler–Lagrange formulation. This formulation is a convenient analytical method for obtaining the dynamic model and studying the physical phenomena of the hexacopter. The Euler–Lagrange formulation uses the total energy of the system confirmed by kinetic and potential energy. The total kinetic energy is defined by the following equation:(7)$$T\left(\dot{\eta }\right)=\frac{1}{2}{\dot{\eta }}^{T}M\;\dot{\eta }$$
where M=diag(m,m,m,I) is a diagonal matrix confirmed by the mass of the hexacopter *m* and the moment of inertia *I* around the vertical axis $B_{z}$. On the other hand, the potential energy is the position or configuration of the system with respect to the world-fixed frame <I>, and the potential energy is described by:(8)$$V\left(\eta \right)=mg\left(\eta_{z}+d_{z}\right)$$
where *g* represents the gravitational acceleration. The Lagrange formulation is obtained by the subtraction of the kinetic $T(\dot{\eta })$ and potential V(η) energy expressed as:(9)$$L=\frac{1}{2}{\dot{\eta }}^{T}M\;\dot{\eta }-mg\left(\eta_{z}+d_{z}\right)$$

Now, applying the Euler–Lagrange formulation:(10)$$\frac{d}{dt}\left(\frac{dL}{\dot{\eta }}\right)-\frac{dL}{\eta }=f_{I}$$
where $f_{I}={\left[\begin{array}{cccc} f_{x_{I}} & f_{y_{I}} & f_{z_{I}} & \tau_{\psi_{I}} \end{array}\right]}^{T}\in R^{4}$ are the generalized forces and torque vector with respect to the frame <I>; with this formulation, it is possible to obtain the nonlinear model by the following equations:(11)$$\left\{\begin{array}{c} m\ddot{\eta_{x}}-d_{y}m\cos\left(\psi \right)\ddot{\eta_{\psi }}-d_{x}m\sin\left(\psi \right)\ddot{\eta_{\psi }}-d_{x}m\cos\left(\psi \right){\dot{\eta_{\psi }}}^{2}+d_{y}m\sin\left(\psi \right){\dot{\eta_{\psi }}}^{2}=f_{x_{I}}\\ m\ddot{\eta_{y}}+d_{x}m\cos\left(\psi \right)\ddot{\eta_{\psi }}-d_{y}m\sin\left(\psi \right)\ddot{\eta_{\psi }}-d_{y}m\cos\left(\psi \right){\dot{\eta_{\psi }}}^{2}-d_{x}m\sin\left(\psi \right){\dot{\eta_{\psi }}}^{2}=f_{y_{I}}\\ m\ddot{\eta_{z}}+mg=f_{z_{I}}\\ I\ddot{\eta_{\psi }}+{d_{x}}^{2}m\;\ddot{\eta_{\psi }}+{d_{y}}^{2}m\;\ddot{\eta_{\psi }}-d_{y}m\cos\left(\psi \right)\ddot{\eta_{x}}+d_{x}m\cos\left(\psi \right)\ddot{\eta_{y}}-d_{x}m\sin\left(\psi \right)\ddot{\eta_{x}}-d_{y}m\sin\left(\psi \right)\ddot{\eta_{y}}=\tau_{\psi_{I}} \end{array}\right.$$
the equations presented in (Equation 11) can be written in a matrix form, resulting in:(12)$$\begin{array}{c} \left[\begin{array}{c} f_{x_{I}}\\ f_{y_{I}}\\ f_{z_{I}}\\ \tau_{\psi_{I}} \end{array}\right]=\left[\begin{array}{cccc} m & 0 & 0 & -m(d_{y}\cos\left(\psi \right)+d_{x}\sin\left(\psi \right))\\ 0 & m & 0 & m(d_{x}\cos\left(\psi \right)-d_{y}\sin\left(\psi \right))\\ 0 & 0 & m & 0\\ -m(d_{y}\cos\left(\psi \right)+d_{x}\sin\left(\psi \right)) & m(d_{x}\cos\left(\psi \right)-d_{y}\sin\left(\psi \right)) & 0 & m{d_{x}}^{2}+m{d_{y}}^{2}+I \end{array}\right]\left[\begin{array}{c} {\ddot{\eta }}_{x}\\ {\ddot{\eta }}_{y}\\ {\ddot{\eta }}_{z}\\ \ddot{\eta_{\psi }} \end{array}\right]\cdots \\ \cdots +\left[\begin{array}{cccc} 0 & 0 & 0 & d_{y}m\sin\left(\psi \right)\dot{\eta_{\psi }}-d_{x}m\cos\left(\psi \right)\dot{\eta_{\psi }}\\ 0 & 0 & 0 & -d_{y}m\cos\left(\psi \right)\dot{\eta_{\psi }}-d_{x}m\sin\left(\psi \right)\dot{\eta_{\psi }}\\ 0 & 0 & 0 & 0\\ 0 & 0 & 0 & 0 \end{array}\right]\left[\begin{array}{c} {\dot{\eta }}_{x}\\ {\dot{\eta }}_{y}\\ {\dot{\eta }}_{z}\\ \dot{\eta_{\psi }} \end{array}\right]+\left[\begin{array}{c} 0\\ 0\\ gm\\ 0 \end{array}\right]\;\;\;\;\;\;\;\;\;\;\;\;\;\;\;\;\;\;\;\;\;\;\;\;\;\;\;\;\; \end{array}$$

To simplify the notation, this work compacts Equation (Equation 12), resulting in the following classical representation:(13)$$f_{I}\left(t\right)=\bar{H}\left(\eta \right)\ddot{\eta }\left(t\right)+\bar{C}\left(\eta ,\dot{\eta }\right)\dot{\eta }\left(t\right)+\bar{g}$$
where $\bar{H}\in R^{4\times 4}$ is the mass and inertia matrix of the hexacopter system, in addition to being positive and symmetric definite; $\bar{C}\in R^{4\times 4}$ is the matrix of Coriolis forces and $\bar{g}\in R^{4}$ is known as the gravitational vector. Vectors $\ddot{\eta }=\left[\begin{array}{cccc} {\ddot{\eta }}_{x} & {\ddot{\eta }}_{y} & {\ddot{\eta }}_{z} & \ddot{\eta_{\psi }} \end{array}\right]$ and $\dot{\eta }=\left[\begin{array}{cccc} {\dot{\eta }}_{x} & {\dot{\eta }}_{y} & {\dot{\eta }}_{z} & \dot{\eta_{\psi }} \end{array}\right]$ are the acceleration and velocity of the point of interest in the inertia frame <I>.

Due to the fact that the hexacopter platform can be controlled through reference maneuverability velocity commands, this work converts the generalized force and torque inputs of the dynamic model (Equation 13) into reference maneuverability velocities as system control inputs. Considering the position dynamic model presented in [54] which considers all DOF in a multirotor:(14)$$\begin{array}{c} \left[\begin{array}{c} \dot{\mu_{l}}\\ \dot{\mu_{m}}\\ \dot{\mu_{n}} \end{array}\right]=\left[\begin{array}{ccc} 0 & w_{z} & -w_{y}\\ -w_{z} & 0 & w_{x}\\ w_{y} & -w_{x} & 0 \end{array}\right]\left[\begin{array}{c} \mu_{l}\\ \mu_{m}\\ \mu_{n} \end{array}\right]-\frac{f_{z_{I}}}{m}\left[\begin{array}{c} 0\\ 0\\ 1 \end{array}\right]+\cdots \\ g\left[\begin{array}{ccc} \cos(\psi )\cos\theta & -\sin\psi \cos\phi +\cos\psi \sin\theta \sin\phi & \sin\psi \sin\phi +\cos\psi \cos\phi \sin\theta \\ \sin(\psi )\cos\theta & \cos\psi \cos\theta +\sin\phi \sin\theta \sin\psi & -\cos\psi \sin\phi +\sin\theta \sin\psi \cos\phi \\ -\sin\theta & \cos\theta \sin\phi & \cos\theta \cos\phi \end{array}\right]\left[\begin{array}{c} 0\\ 0\\ 1 \end{array}\right] \end{array}$$
it can be written in a compact form as $\dot{\mu }={\left[\begin{array}{c} w \end{array}\right]}_{x}\mu +gR{\left(\theta ,\phi ,\psi \right)}^{T}e_{3}-\frac{f_{z}}{m}e_{3}$, where $w=\left[\begin{array}{ccc} w_{x} & w_{y} & w_{z} \end{array}\right]$ are the angular velocities associated with the frame <B>, ${\left[\begin{array}{c} w \end{array}\right]}_{x}$ represents skew symmetric matrix and finally R(θ,ϕ,ψ) represents the rotation matrix from the frame <B> to <I>. For any multicopter, drags applied to rotating blades are in the direction of the body axes; from (Equation 14), the position dynamic model considering the aerodynamic drag model is represented as:(15)$$\left[\begin{array}{c} \dot{\mu_{l}}\\ \dot{\mu_{m}} \end{array}\right]=\left[\begin{array}{c} \mu_{m}\omega_{z}-\mu_{n}\omega_{y}-g\sin\theta -\frac{k_{drag}}{m}\mu_{l}\\ \mu_{n}\omega_{x}-\mu_{l}\omega_{z}+g\cos\theta \sin\phi -\frac{k_{drag}}{m}\mu_{m} \end{array}\right]$$
where $\dot{\mu_{l}}$, $\dot{\mu_{m}}$ are the accelerations in the body frame <B> and $k_{drag}$ is the drag constant. To include the drag coefficients in the dynamic model, this work split Equation (Equation 14) and can be written as:(16)$$\left[\begin{array}{c} a_{x}\\ a_{y} \end{array}\right]=\left[\begin{array}{c} \mu_{m}\omega_{z}-\mu_{n}\omega_{y}-\frac{k_{drag}}{m}\mu_{l}\\ \mu_{n}\omega_{x}-\mu_{l}\omega_{z}-\frac{k_{drag}}{m}\mu_{m} \end{array}\right]$$
where $a_{x}=\dot{\mu_{l}}+g\sin\left(\theta \right)$ and $a_{y}=\dot{\mu_{m}}-g\cos\left(\theta \right)\sin\left(\phi \right)$. Since the low-level controller guarantees hover, the cross terms are ignored, so the specific force satisfies:(17)$$\left[\begin{array}{c} f_{x}\\ f_{y} \end{array}\right]=\left[\begin{array}{c} -k_{drag}\mu_{l}\\ -k_{drag}\mu_{m} \end{array}\right]$$
where the forces produced in the system $f_{hB}=\left[\begin{array}{cc} f_{x} & f_{y} \end{array}\right]$ are generate by a PD controller for the horizontal plane. The PD controller was designed as:(18)$$f_{hB}=k_{pl}\left(\mu_{h_{ref}}-\mu_{h}\right)+k_{dl}\left({\dot{\mu }}_{h_{ref}}-{\dot{\mu }}_{h}\right)$$
where $\mu_{h}={\left[\begin{array}{cc} \mu_{l} & \mu_{m} \end{array}\right]}^{T}\in R^{2}$ is the vector of the horizontal maneuverability velocities, $\mu_{h_{ref}}={\left[\begin{array}{cc} \mu_{lref} & \mu_{mref} \end{array}\right]}^{T}\in R^{2}$ is the reference desired velocity and $k_{pl}$, $k_{dl}$ are positive definite matrices. For the PD controller, this work used the following assumptions if ${\dot{\mu }}_{h_{ref}}=0$ then $\lim_{t\to \infty }∥f_{h}∥=0$ and $\lim_{t\to \infty }∥{\tilde{\mu }}_{h}∥=0$, respectively.

A similar concept was applied to the vertical force attached to the body frame $f_{z_{B}}=f_{ref}+mg$. Designing a PD controller for the vertical thrust as:(19)$$f_{z_{B}}=k_{pn}\left(\mu_{n_{ref}}-\mu_{n}\right)+k_{dn}\left({\dot{\mu }}_{n_{ref}}-{\dot{\mu }}_{n}\right)+mg$$
where $k_{pn}$, $k_{dn}$ are positive definite matrices, this work considers the following assumptions if ${\dot{\mu }}_{n_{ref}}=0$ then the $\lim_{t\to \infty }∥f_{z_{I}}∥=0$ and $\lim_{t\to \infty }∥{\tilde{\mu }}_{n}∥=0$, respectively.

The same approach can be extended to the attitude dynamic model presented in [54], which is defined as:(20)$$J_{B}\cdot \dot{w}=-w\times \left(J_{B}\cdot w\right)+G_{B}+\tau$$
where $\tau ≜\left[\begin{array}{ccc} \tau_{x} & \tau_{y} & \tau_{\psi } \end{array}\right]\in R^{3}$ is the vector of moments in the body axes <B>; $J_{B}\in R^{3\times 3}$ includes the moment of inertia and $G_{B}≜{\left[\begin{array}{ccc} G_{B,\phi } & G_{B,\theta } & G_{B,\psi } \end{array}\right]}^{T}\in R^{3}$ represents the gyroscopic torques. Ignoring the term $-w\times \left(J_{B}\cdot w\right)+G_{B}$, the simplified attitude dynamic model is defined as: $J_{B}\dot{w}=\tau_{\psi }$, where a PD controller for angular translations is defined as:(21)$$\tau_{\psi }=k_{p\omega }\left(\omega_{ref}-\omega \right)+k_{d\omega }\left({\dot{\omega }}_{ref}-\dot{\omega }\right)$$
where $k_{p\omega },k_{d\omega }\in R$ are positive scalars and the following assumptions are true if ${\dot{\omega }}_{ref}=0$; then, $\lim_{t\to \infty }∥\tau_{\psi }∥=0$ and $\lim_{t\to \infty }∥\tilde{\omega }∥=0$, respectively.

After all the considerations presented beforehand, this work introduces the general structure of low-level PD controllers combining Equations (Equation 18), (Equation 19) and (Equation 21), finally the structure is presented in Equation (Equation 22).
(22)$$\left[\begin{array}{c} f_{x_{B}}\\ f_{y_{B}}\\ f_{z_{B}}\\ \tau_{\psi_{B}} \end{array}\right]=\left[\begin{array}{cccc} k_{pl} & 0 & 0 & 0\\ 0 & k_{pl} & 0 & 0\\ 0 & 0 & k_{pn} & 0\\ 0 & 0 & 0 & k_{p\omega } \end{array}\right]\left[\begin{array}{c} \mu_{lref}\\ \mu_{mref}\\ \mu_{nref}\\ \omega_{ref} \end{array}\right]-\left[\begin{array}{cccc} k_{pl} & 0 & 0 & 0\\ 0 & k_{pl} & 0 & 0\\ 0 & 0 & k_{pn} & 0\\ 0 & 0 & 0 & k_{p\omega } \end{array}\right]\left[\begin{array}{c} \mu_{l}\\ \mu_{m}\\ \mu_{n}\\ \omega \end{array}\right]-\left[\begin{array}{cccc} k_{dl} & 0 & 0 & 0\\ 0 & k_{dl} & 0 & 0\\ 0 & 0 & k_{dn} & 0\\ 0 & 0 & 0 & k_{d\omega } \end{array}\right]\left[\begin{array}{c} {\dot{\mu }}_{l}\\ {\dot{\mu }}_{m}\\ {\dot{\mu }}_{n}\\ \dot{\omega } \end{array}\right]+\left[\begin{array}{c} 0\\ 0\\ mg\\ 0 \end{array}\right]$$

Equation (Equation 22) can be written in a compact form as $f_{B}\left(t\right)=K_{p}\mu_{\mathrm{ref}}\left(t\right)-K_{p}\mu \left(t\right)-K_{d}\dot{\mu }\left(t\right)+g$, where the values $k_{pl}$, $k_{pm}$ and $k_{p\omega }$ are the proportional values of the PD controller; $k_{dl}$, $k_{dm}$ and $k_{d\omega }$ are the derivative gains; $\mu_{\mathrm{ref}}={\left[\begin{array}{cccc} \mu_{lref} & \mu_{mref} & \mu_{nref} & \omega_{ref} \end{array}\right]}^{T}\in R^{4}$ are the reference maneuverability velocities or control vector of the hexacopter; μ and $\dot{\mu }$ are the real velocities and accelerations generated in the system.

Finally, with all the transformations considered beforehand, Equations (Equation 4) and (Equation 6) are substituted into a dynamic hexarotor model (Equation 12), and equating the expression to (Equation 22), a new reduced dynamic model can written as:(23)$$\begin{array}{c} \left[\begin{array}{c} \mu_{lref}\\ \mu_{mref}\\ \mu_{nref}\\ \omega_{ref} \end{array}\right]=\left[\begin{array}{cccc} \zeta_{1} & 0 & 0 & \zeta_{2}\\ 0 & \zeta_{3} & 0 & \zeta_{4}\\ 0 & 0 & \zeta_{5} & 0\\ b\zeta_{6} & a\zeta_{7} & 0 & \zeta_{8}\left(a^{2}+b^{2}\right)+\zeta_{9} \end{array}\right]\left[\begin{array}{c} {\dot{\mu }}_{l}\\ {\dot{\mu }}_{m}\\ {\dot{\mu }}_{n}\\ \dot{\omega } \end{array}\right]+\left[\begin{array}{cccc} \zeta_{10} & \omega \zeta_{11} & 0 & a\omega \zeta_{12}\\ \omega \zeta_{13} & \zeta_{14} & 0 & b\omega \zeta_{15}\\ 0 & 0 & \zeta_{16} & 0\\ a\omega \zeta_{17} & b\omega \zeta_{18} & 0 & \zeta_{19} \end{array}\right]\left[\begin{array}{c} \mu_{l}\\ \mu_{m}\\ \mu_{n}\\ \omega \end{array}\right] \end{array}$$

This new formulation represents the evolution of the hexacopter’s general velocities with the reference maneuverability velocities as the input of the system. This representation is useful because commercial aerial platforms can be directly controlled with maneuverability velocities, ignoring the low-level control. Equation (Equation 23) can be compactly written as:(24)$$\mu_{\mathrm{ref}}\left(t\right)=H\left(\zeta ,a,b\right)\dot{\mu }\left(t\right)+C\left(\zeta ,\mu \right)\mu \left(t\right)$$
where $H={\left({\mathrm{RK}}_{p}\right)}^{-1}((\bar{H}J+RK_{d})$, $C={\left({\mathrm{RK}}_{p}\right)}^{-1}\left(\bar{H}\dot{J}+RK_{p}+\bar{C}J\right)$, $G={\left({\mathrm{RK}}_{p}\right)}^{-1}\left(g-R\bar{G}\right))=0$. Thus, $H\left(\zeta ,a,b\right)\in R^{4\times 4}$ is a positive definite matrix, which is the new mass and inertia matrix of the hexacopter robot, the new Coriolis and Centripetal matrix is defined by $C\left(\zeta ,\mu \right)\in R^{4\times 4}$, R represents the rotation matrix under the z axis from frame <B> to frame <I>. The vector of dynamic parameters represents the combination of all the internal values in the hexacopter robot such as the physical, mechanical, electrical and PD values, and the vector is defined by $\zeta =\left[\begin{array}{cccc} \zeta_{1} & \zeta_{2} & \ldots & \zeta_{l_{d}} \end{array}\right]\in R^{l_{d}}$ where $l_{d}=19$ is the number of variables that this work needs to identify as the details of each parameters are presented in Appendix A for Equations (Equation 46) and (Equation 47).

##### Euler–Lagrange Identification and Validation

This section presents the identification and validation of the dynamic parameters, as this work uses real experimental data to estimate these values in order to use the model in the proposed controller. This work transforms the mathematical model (Equation 24) in the linear parametric representation into:(25)$$\mu_{\mathrm{ref}}\left(t\right)=\Theta \left(\dot{\mu }\left(t\right),\mu \left(t\right)\right)\zeta$$
where the matrix Θ confirms the values of velocities and accelerations obtained from real-world experiments. In order to estimate the vector of dynamic values, this work uses snapshot measurements over the time as $l\in [t,t+t_{f}]$ where *l* is an instant of measure and $t_{f}$ is the final time of the experimental information. The identification process was developed through optimization techniques defining $\ell_{m}\left(\dot{\mu }\left(l|t\right),\mu \left(l|t\right),\mu_{\mathrm{ref}}\left(l|t\right),\zeta \right):R^{4}\to R_{\geq 0}$, which is a positive semi-definite function confirmed by:(26)$$\ell_{m}\left(\dot{\mu }\left(l|t\right),\mu \left(l|t\right),\mu_{\mathrm{ref}}\left(l|t\right),\zeta \right)=\frac{1}{2}\overset{Model\;identification\;cost}{\overset{︷}{\left|\right|\mu_{\mathrm{ref}}\left(l|t\right)-\Theta \left(\dot{\mu }\left(l|t\right),\mu \left(l|t\right)\right)\zeta {\left|\right|}_{Q_{m}}^{2}}}$$
*Model identification cost:* this function was formulated using the subtraction between the reference maneuverability $\mu_{\mathrm{ref}}\left(l|t\right)$ velocities applied in the hexacopter and the system measurements defined by the matrix $\Theta (\dot{\mu }\left(l|t\right),\mu \left(l|t\right))$; the operator ||·|| is known as the Euclidean norm; and $Q_{m}\in R_{>0}^{4\times 4}$ is a positive definite weighting matrix.

With Equation (Equation 26), it is possible to generate the functional cost or performance index over the experimental data, which is formulated as follows:(27)$$V_{m}\left(\dot{\mu }\left(l|t\right),\mu \left(l|t\right),\mu_{\mathrm{ref}}\left(l|t\right),\zeta \right)=\int_{t}^{t+t_{f}}\ell_{m}\left(\dot{\mu }\left(\tau |t\right),\mu \left(\tau |t\right),\mu_{\mathrm{ref}}\left(\tau |t\right),\zeta \right)d\tau$$

Considering the performance index function presented in (Equation 27), the optimization structures are defined in (Equation 28), which was solved using the sequential quadratic programming (SQP) technique.
(28)$$P_{m}:P_{m}\left(\dot{\mu }\left(t\right),\mu \left(t\right),\mu_{\mathrm{ref}}\left(t\right)\right)=\min_{\zeta }\;V_{m}\left(\dot{\mu }\left(l|t\right),\mu \left(l|t\right),\mu_{\mathrm{ref}}\left(l|t\right),\zeta \right)$$

This work used the experimental information of the hexacopter platform DJI MATRICE 600 PRO to identify the dynamic parameters. The aerial system has a low-level flight controller called A3, which is confirmed by three global positioning systems (GPS) and three inertial measurement unit systems (IMUs). Furthermore, this system provides highly precise measurements from the system states. The identified parameters are presented in Table 1; furthermore, for the identification and validation process, this work used a different kind of reference velocity signals.

Figure 2 shows the results of the identification process, where the signals $\mu_{l}$, $\mu_{m}$, $\mu_{n}$ and ω were obtained by real-world experiments; and $\mu_{lm}$, $\mu_{mm}$, $\mu_{nm}$ and $\omega_{m}$ are the estimated values of the dynamic model using the optimization formulation presented previously. The presence of noise in the signals is an important factor due to the fact that it probably causes problems in the identification process; this work uses a filter defined as: λ/(s+λ) with λ=1 to eliminate noise in the real measurements. The filter was applied to the system measurements and the reference maneuverability velocities.

The validation process of the dynamic parameters is a very important factor; hence, the model needs to estimate the behavior of the system with different signals in order to represent the real dynamics of the system. The validation signals are shown in Figure 3, the validation process shows the performance of the estimate dynamic model, which will be useful for the comparative section.

With all the previously presented results and the validation of the dynamic parameters, this work formulates the identified model using Euler-Lagrange as a vector-function presented as:(29)$$\dot{\mu }\left(t\right)=f_{SQP}\left(\zeta ,\mu \left(t\right),\mu_{\mathrm{ref}}\left(t\right)\right)$$

##### Dynamic Mode Decomposition Formulation

This section presents the second approach of the dynamic model formulation which, due to the hexacopter platform, only has internal low-level PIDs; and the system tries to stay close to the hover-position with small angular variations in roll and pitch angles (ϕ,θ). Furthermore, this kind of hexacopter platforms cannot be controlled with the torques commands, and the control inputs at the system are the reference maneuverability velocities $\mu_{\mathrm{ref}}$. Considering the small angle assumptions and the inputs in the system, the model can hence be formulated with a linear approximation for time-varying systems resulting in:(30)$$\dot{\mu }\left(t\right)=A\mu \left(t\right)+B\mu_{\mathrm{ref}}\left(t\right)$$
where μ and $\dot{\mu }$ are the vectors of the maneuverability velocities and accelerations; $\mu_{\mathrm{ref}}$ is the reference maneuverability velocity vector also known as the control vector; matrices $A\in R^{4\times 4}$ and $B\in R^{4\times 4}$ represent the unforced system and the contribution of the control vector, respectively.

One of the emerging techniques to identify systems by experimental information is that of dynamic mode decomposition (DMD) [55,56,57], where the objective is to approximate the matrices A and B. The DMD algorithm proposes the construction of snapshot measurements *s* and the formation of augmented matrices, resulting in a new formulation of the system as follows:(31)$$\dot{\chi }\approx A\chi +B\Gamma$$
where the new matrices are defined below:$$\dot{\chi }=\left[\begin{array}{cccc} | & | & & |\\ \dot{\mu_{1}} & \dot{\mu_{2}} & \cdots & \dot{\mu_{s}}\\ | & | & & | \end{array}\right]$$
$$\chi =\left[\begin{array}{cccc} | & | & & |\\ \mu_{1} & \mu_{2} & \cdots & \mu_{s}\\ | & | & & | \end{array}\right]$$
$$\Gamma =\left[\begin{array}{cccc} | & | & & |\\ \mu_{{\mathrm{ref}}_{1}} & \mu_{{\mathrm{ref}}_{2}} & \cdots & \mu_{{\mathrm{ref}}_{s}}\\ | & | & & | \end{array}\right]$$
$\dot{\chi }$ and χ are the augmented matrices confirmed by the maneuverability accelerations and velocities, respectively; and Γ is the matrix of the reference velocities applied in the system over the *s* number of snapshots.

For estimation purposes, the system can be formulated as:(32)$$\dot{\chi }\approx \Phi \Omega$$
where $\Phi =\left[\begin{array}{cc} A & B \end{array}\right]\in R^{4\times (4+4)}$ is the matrix confirmed by unknown values of A and B; and $\Omega ={\left[\begin{array}{cc} \chi & \Gamma \end{array}\right]}^{T}\in R^{(4+4)\times s}$ is the data matrix obtained from the system. This work finds the best values of the unknown matrix using the Frobenius norm $\left|\right|\dot{\chi }-\Phi \Omega {\left|\right|}_{F}$, defining the values in the following equation:(33)$$\Phi \approx \dot{\chi }\Omega^{†}$$

One of the best methods to find the solution of the Frobenius norm is the singular value decomposition (SVD); therefore, applying this method to the augmented data matrix resulting as $\Omega \approx \bar{U}\bar{\Sigma }{\bar{V}}^{T}$, where $\bar{U}$, $\bar{\Sigma }$ and $\bar{V}$ are matrices with a truncation value $\bar{r}$, with these considerations presented above, the estimation is defined by the following equation:(34)$$\Phi \approx \dot{\chi }\bar{V}{\bar{\Sigma }}^{-1}{\bar{U}}^{T}$$

To approximate the values of the matrices A and B, the Equation (Equation 34) can be written as (Equation 35), considering the splitting values of the singular vector $\bar{U}$.
(35)$$\left[\begin{array}{cc} \bar{A} & \bar{B} \end{array}\right]\approx \left[\begin{array}{cc} \dot{\chi }\bar{V}{\bar{\Sigma }}^{-1}{\bar{U}}_{1}^{T} & \dot{\chi }\bar{V}{\bar{\Sigma }}^{-1}{\bar{U}}_{2}^{T} \end{array}\right]$$

The values of the matrix A can be approximated using the split matrix ${\bar{U}}_{1}\in R^{4\times \bar{r}}$ and the values of B are approximated using the values of ${\bar{U}}_{2}\in R^{4\times \bar{r}}$.

##### Dynamic Mode Decomposition Identification and Validation

This section presents the identification and validation process of the dynamic model using the DMD formulation. The values of matrices $\dot{\chi }$, χ and Γ were constructed by measure snapshots information of the aerial platform DJI MATRICE 600 PRO and the application of Algorithm 1, which represents all the necessary steps to identify the values of unknown matrices.

**Algorithm 1** Identification process Using DMD**Input:** DJI MATRICE 600 PRO measurements $\left(\dot{\chi },\chi ,\Gamma \right)$ and truncation value $\bar{r}$.
**Output:** Matrices $\bar{A}$ and $\bar{B}$.
    **Ident**
$\left(\dot{\chi },\chi ,\Gamma ,\bar{r}\right)$
**do**
     $\Omega \leftarrow {\left[\begin{array}{cc} \chi & \Upsilon \end{array}\right]}^{T}$
     (U,Σ,V)←SVD(Ω)
     $\left(\bar{U},\bar{\Sigma },\bar{V}\right)\leftarrow Trunc\left(U,\Sigma ,V,\bar{r}\right)$
     $\left({\bar{U}}_{1},{\bar{U}}_{2}\right)\leftarrow Split\left(\bar{U}\right)$
     $\bar{A}\leftarrow \dot{\chi }\bar{V}{\bar{\Sigma }}^{-1}{\bar{U}}_{1}^{T}$
     $\bar{B}\leftarrow \dot{\chi }\bar{V}{\bar{\Sigma }}^{-1}{\bar{U}}_{2}^{T}$
    **end Ident**
    **return**
$\left(\bar{A},\bar{B}\right)$


The approximation values of the unknown matrix are presented in Table 2 and Table 3, where, $a_{ij}$ and $b_{ij}$ are the individual elements of $\bar{A}$ and $\bar{B}$, respectively.

The signals used for the identification process are presented in Figure 4, which are the same as those used for the identification process employing Euler–Lagrange formulation, due to the fact that both techniques need to estimate the values under the same conditions.

The estimated model displays good behavior due to the fact it replicates the signals obtained from experimental data. The noise in the angular maneuverability velocity ω is an important factor, however, dynamic mode decomposition and the robustness of the Frobenius norm can deal with this noise and the application of filters is not necessary in the identification process.

In the same way presented in the validation process using the Lagrange formulation, this section presents the validation process under DMD formulation, the results of which are shown in Figure 5, which shows the good performance of the proposed technique.

Finally, considering the approximation values of the unknown matrices, the dynamic model can be written in a vector-function form as:(36)$$\dot{\mu }\left(t\right)=f_{DMD}\left(\Gamma ,\mu \left(t\right),\mu_{\mathrm{ref}}\left(t\right)\right)$$

#### 2.1.3. Accuracy Comparative Results

This section presents the comparative results between the Euler–Lagrange (Equation 29) and dynamic mode decomposition (Equation 36) formulations. This work uses the integral square error (ISE) to show the performance of each formulation and selects the best one for the nonlinear model predictive controller structure.

Figure 6 shows the ISE of each formulation, where the model for frontal maneuverability velocity $\mu_{lm}$ presents a considerable error using the Euler–Lagrange formulation—with a value of precisely 1.26—whilst on the other hand, DMD results show a better performance with a value of 0.56. The proposed formulations show similar results with respect to the lateral velocity $\mu_{mm}$, and the values using the Euler–Lagrange and DMD are 0.35 and 0.16, respectively. The results for the estimated upper maneuverability velocity $\mu_{nm}$ are really close with the following values 0.0104 and 0.0100. Finally, the ISE of the estimated models for the angular maneuverability velocity are 0.104 and 0.038 for the Euler–Lagrange and DMD formulations, respectively.

With the previously presented results, this work uses the DMD estimated model to construct the proposed controller. Combining Equations (Equation 5) and (Equation 36), the general mathematical representation is defined as:(37)$$\dot{x}\left(t\right)=f\left(x\left(t\right),\mu_{\mathrm{ref}}\left(t\right)\right)$$
$$f\left(x\left(t\right),\mu_{\mathrm{ref}}\left(t\right)\right)\;:=\left[\begin{array}{c} f_{k}\left(\psi \left(t\right),\mu \left(t\right)\right)\\ f_{DMD}\left(\Gamma ,\mu \left(t\right),\mu_{\mathrm{ref}}\left(t\right)\right) \end{array}\right]$$
where $x={\left[\begin{array}{cc} \eta^{T} & \mu^{T} \end{array}\right]}^{T}\in R^{8}$ is the generalized vector of the system states that is confirmed by η, which is the pose of the point of interest in the hexacopter and μ is the maneuverability velocities.

### 2.2. Control Methodology

This section presents the formulation of the proposed controller, where the main objective is the trajectory tracking over the desired reference trajectory using an hexacopter platform and due to the dynamic effects and the important size of the system, this work includes obstacle avoidance to improve the behavior of the system in dynamic environments. This work presents the nonlinear model predictive controller (NMPC); furthermore, the constraints included in the optimization problem were: generalized system dynamics, bounded maneuverability velocities, rate of change of control velocities and obstacles in the dynamic environment. The main objectives of the proposed controller are: to solve the control problem associated with the trajectory tracking subject to the system and the environment and system constraints. The control problem is solved by the scheme shown in Figure 7, where the structure uses the system information and possible obstacles in the environment to generate an optimal control policy under the system constraints.

#### 2.2.1. Nonlinear Model Predictive Control

In order to solve the NMPC proposed scheme, this work formulates the generalized dynamics (Equation 37) as a prediction, resulting in:(38)$$\dot{x}\left(l|t\right)=f\left(x\left(l|t\right),\mu_{\mathrm{ref}}\left(l|t\right)\right)$$
where l∈[t,t+T] is the instant value evolution between the initial time *t* and the prediction horizon *T*. Due to the prediction behavior of this scheme, the NMPC formulation requires index performances over the prediction horizon *T* that guarantees the solution of the control problem and the system constraints. The generation of the performance functions are presented below.

#### 2.2.2. Tracking Trajectory Formulation

This subsection presents the generation of the cost function over the evolution instants *l* considering the following property: $\ell_{t}\left(x\left(l|t\right),\eta_{\mathrm{ref}}\left(l|t\right),\mu_{\mathrm{ref}}\left(l|t\right)\right):R^{4}\times R^{4}\to R_{\geq 0}$ to formulate the function as:(39)$$\ell_{t}\left(x\left(l|t\right),\eta_{\mathrm{ref}}\left(l|t\right),\mu_{\mathrm{ref}}\left(l|t\right)\right)=\overset{Trajectory\;stage\;cost}{\overset{︷}{\frac{1}{2}\left|\right|\eta_{\mathrm{ref}}\left(l|t\right)-W_{t}x\left(l|t\right){\left|\right|}_{Q_{t}}^{2}}}+\underset{Control\;stage\;cost}{\underset{︸}{\frac{1}{2}\left|\right|\mu_{\mathrm{ref}}\left(l|t\right){\left|\right|}_{Q_{u}}^{2}}}$$

*Trajectory Stage Cost:* this part of the function is structured by the control error between the desired trajectory $\eta_{\mathrm{ref}}={\left[\begin{array}{cccc} \eta_{x_{ref}} & \eta_{y_{ref}} & \eta_{z_{ref}} & \eta_{\psi_{ref}} \end{array}\right]}^{T}\in R^{4}$ and the measurements of the system x; the constant matrix $W_{t}\in R^{4\times 8}$ produces linear mapping between all the states of the system and the outputs $\eta_{x}$, $\eta_{y}$, $\eta_{z}$ and $\eta_{\psi }$; the Euclidean norm is defined by the operator ||·||; finally, $Q_{t}\in R_{>0}^{4\times 4}$ is a constant positive definite matrix.

*Control stage cost:* this part of the cost function defines the variations between the maneuverability control velocities $\mu_{\mathrm{ref}}\left(l|t\right))$ considering that $Q_{u}\in R_{>0}^{4\times 4}$ is a positive definite weighting matrix.

The cost function at the last step of the prediction horizon *T* was formulated with following property: $\ell_{T}\left(x\left(t+T|t\right),\eta_{\mathrm{ref}}\left(t+T|t\right)\right):R^{4}\to R_{\geq 0}$; with this consideration, the cost function is defined as:(40)$$\ell_{T}\left(x\left(t+T|t\right),\eta_{\mathrm{ref}}\left(t+T|t\right)\right)=\overset{Terminal\;Trajectory\;cost}{\overset{︷}{\frac{1}{2}\left|\right|\eta_{\mathrm{ref}}\left(t+T|t\right)-Wx\left(t+T|t\right){\left|\right|}_{Q_{T}}^{2}}}$$
*Terminal trajectory cost:* this term considers the control error between the reference trajectory at the last step of the prediction horizon *T*, where $Q_{T}\in R_{>0}^{4\times 4}$ is a positive definite weighting matrix.

This work uses Equations (Equation 39) and (Equation 40) to generate the performance index over the prediction horizon, which was developed in the continuous time formulation, where τ is the interval analyzed in the integral operation, and the performance index is defined as:(41)$$V\left(x\left(l|t\right),\eta_{\mathrm{ref}}\left(l|t\right),\mu_{\mathrm{ref}}\left(l|t\right)\right)=\int_{t}^{t+T}\;\ell_{t}\left(x\left(\tau |t\right),\eta_{\mathrm{ref}}\left(\tau |t\right),\mu_{\mathrm{ref}}\left(\tau |t\right)\right)d\tau +\ell_{T}\left(x\left(t+T|t\right),\eta_{\mathrm{ref}}\left(t+T|t\right)\right)$$

The performance index function (Equation 41) guarantees the trajectory tracking propriety of the controller. The tuning process of the positive definite matrices were developed in the simulations in order to prevent accidents in real-world experiments.

#### 2.2.3. Maneuverability Velocities Constraints

Due to the hexacopter platform having maximum maneuverability velocities, this work proposes control input constraints on the NMPC structure, in addition to further constraints on the successive difference of the maneuverability velocities which were included to prevent aggressive behaviors. The input constraints are defined in (Equation 42) in order to maintain the control signals generated by the proposed structure under the limits of the hexacopter platform
(42)$$U=\{\mu_{\mathrm{ref}}\in R^{4}:\mu_{{\mathrm{ref}}_{min}}\leq \mu_{\mathrm{ref}}\leq \mu_{{\mathrm{ref}}_{max}};|\mu_{\mathrm{ref}}\left(j+1|t\right)-\mu_{\mathrm{ref}}\left(j|t\right)|\leq \Delta \mu_{{\mathrm{ref}}_{max}}\}$$
where $\mu_{{\mathrm{ref}}_{min}}\in R^{4}$ and $\mu_{{\mathrm{ref}}_{max}}\in R^{4}$ are the vectors of lower and upper limits in maneuverability velocities; to the successive difference, this works considers j∈[t,t+(T−1)] and $\Delta \mu_{{\mathrm{ref}}_{max}}$ as the vector of the maximum rate of change between the maneuverability velocities.

#### 2.2.4. Obstacle Constraints

This section presents the obstacle constraints in order to track the desired trajectory and avoid obstacles in the dynamic environments. The obstacles can be defined as $\xi^{obs}\left(i|z\right)={\left[\begin{array}{ccc} \eta_{x}^{obs} & \eta_{y}^{obs} & \eta_{z}^{obs} \end{array}\right]}^{T}\in R^{3}$ like a position vector with respect to the inertia frame <I>, where i∈[z,z+Z] considering the analyzed obstacle as *z* and *Z* the number of obstacles detected by the system, the representation of the obstacles is shown in Figure 8.

The function of the distance between obstacles and the hexacopter has the following property: $\ell_{obs}\left(x\left(l|t\right),\xi^{obs}\left(i|z\right)\right):R^{4}\to R_{\geq 0}$, and the equation is formulated as:(43)$$\ell_{obs}\left(x\left(l|t\right),\xi^{obs}\left(i|z\right)\right)=\overset{Distance\;to\;Obstacles}{\overset{︷}{\left|\right|\xi^{obs}\left(i|z\right)-W_{o}x\left(l|t\right){\left|\right|}^{2}}}$$
where $W_{o}\in R^{3x8}$ is a constant matrix that produces linear mapping between all the states of the system and the outputs $\eta_{x}$, $\eta_{y}$ and $\eta_{z}$. With the previously presented considerations, this work formulates the system states constraints in (Equation 44), and where the safe region was defined as $r_{obs}\in R_{>0}$, this value includes the safely region and the rotating blades’ positions.
(44)$$O=\{x\in R^{4}:r_{obs}-\ell_{obs}\left(x\left(l|t\right),\xi^{obs}\left(i|z\right)\right)\leq 0\}$$

#### 2.2.5. Nonlinear Model Predictive Control Formulation

Considering the performance index presented before (Equation 41) and the system states (Equation 44) and control (Equation 42) constraints, this work can formulate the optimization problem as:
(45a)$$\begin{array}{c} P:P\left(\eta_{\mathrm{ref}}\left(t\right)\right)=\min_{\mu_{\mathrm{ref}}\left(l|t\right),x\left(l|t\right)}\;\int_{t}^{t+T}\;\ell_{t}\left(x\left(\tau |t\right),\eta_{\mathrm{ref}}\left(\tau |t\right),\mu_{\mathrm{ref}}\left(\tau |t\right)\right)d\tau +\ell_{T}\left(x\left(t+T|t\right),\eta_{\mathrm{ref}}\left(t+T|t\right)\right) \end{array}$$
(45b)$$\begin{array}{c} \mathrm{subject}\mathrm{to}:\dot{x}\left(l|t\right)-f\left(x\left(l|t\right),\mu_{\mathrm{ref}}\left(l|t\right)\right)=0\;\;\;\;\;\;\;\;\;\;\;\;\;\;\;\;\;\;\;\;\;\;\; \end{array}$$ with the following constrains: (45c)$$\begin{array}{c} x\left(l|t\right)\in r_{obs}-\ell_{obs}\left(x\left(l|t\right),\xi^{obs}\left(i|z\right)\right)\leq 0\;\;\;\;\;\;\;\;\;\;\;\;\;\;\;\;\;\;\;\;\;\;\;\;\;\;\;\;\;\;\;\; \end{array}$$
(45d)$$\begin{array}{c} \mu_{\mathrm{ref}}\left(l|t\right)\in \mu_{{\mathrm{ref}}_{min}}\leq \mu_{\mathrm{ref}}\leq \mu_{{\mathrm{ref}}_{max}};\left|\mu_{\mathrm{ref}}\left(j+1|t\right)-\mu_{\mathrm{ref}}\left(j|t\right)\right|\leq \Delta \mu_{{\mathrm{ref}}_{max}}\;\;\;\;\;\;\; \end{array}$$
P is the function that guarantees the correct trajectory tracking and obstacle avoidance of the hexacopter system. The optimal control problem ([Disp-formula FD45a-sensors-22-04712]) was converted into a nonlinear programming formulation (NLP) using the direct multiple shooting method, where the control maneuverability velocities $\mu_{\mathrm{ref}}\left(l|t\right)$ and the system states x(l|t) are the decision variables of the optimization problem. A multiple shooting technique is more computationally efficient when it is compared with other discretization formulation, e.g., single shooting, more information in [58]. The system model was considered as the optimization constrain defined by (45b), the system and control constraints formulated in (45c) and (45d), respectively. With these considerations, this work minimizes the nonlinear programming problem using CasADI as a nonlinear optimization framework [59].

## 3. Results and Discussions

This section presents the simulations and experimental results to validate the proposed controller for the hexacopter platform DJI MATRICE 600 PRO. The experiments were performed considering: *(i) Real-world experiment:* the results presented in this section were developed using the nonlinear model predictive controller and environment obstacles are not considered in this experiments due to the hexacopter platform not having the required sensor for real-world measurements. *(ii) Simulation experiment:* simulation experiments show the effectiveness of the proposed controller in highly dynamic simulation environments with simulations of the dynamic behavior of the hexacopter in order to make it more realistic and prove the scalability of the controller.

### 3.1. Real-World Experiments

Several experiments on trajectory tracing were performed in order to demonstrate the performance of the proposed controller. In order to demonstrate the effectiveness of the optimization structure, the controller was implemented at onboard computer of the hexacopter through the software Matlab, additionally the hexacopter platform is shown in Figure 9 with the following hardware features Intel processor i7-7700HQ and CPU 2.80 GHz × 8. The controller was developed considering the optimal control problem ([Disp-formula FD45a-sensors-22-04712]), the evolution of the identified model was developed using the fourth-order Runge–Kutta method, with a sample time of $t_{s}=0.1\left[s\right]$ and the final time of the experiment is defined as $t_{f}=100\left[s\right]$.

The experiment was performed in the city of Ambato, in the province of Tungurahua, Ecuador, and started at 09:36 on 12 January 2022. The wind velocity at the time of the experiment was approximately 10.1 km/h, as shown in Figure 10.

The desired reference trajectory for the hexacopter and the initial conditions are defined in Table 4, where $\left(\eta_{x_{o}},\eta_{y_{o}},\eta_{z_{o}},\eta_{\psi_{o}}\right)$ and $\left(\mu_{l_{o}},\mu_{m_{o}},\mu_{n_{o}},\omega_{o}\right)$ are the initial position and maneuverability velocities of the hexacopter, respectively.

To implement the proposed controller, the constant values of positive definite matrices are defined in Table 5, and were obtained through a variety of experiments in order to improve the performance of the controller scheme.

The results of the experiment are illustrated in Figure 11, Figure 12, Figure 13 and Figure 14. Figure 11 shows the movement of the hexacopter platform based on real information over the experiment. The hexacopter tracks the desired reference trajectory; however, the results present a small control error in the trajectory due to the wind velocity that acts as an external disturbance. The reference trajectory is defined as $\eta_{\mathrm{ref}}=\left[\begin{array}{cccc} \eta_{x_{ref}} & \eta_{y_{ref}} & \eta_{z_{ref}} & \eta_{\psi_{ref}} \end{array}\right]$ and the hexacopter system states during the experiment were confirmed by η.

The previously presented results show the performance of the controller that guarantees the execution of the reference trajectory over different axes.

The control signals represented in Figure 12 are generated by the proposed control scheme, the external disturbances produce a miss match in the model but the controller generates the adequate smooth control actions.

Figure 13 shows the control errors of the proposed controller, where ${\tilde{\eta }}_{x},{\tilde{\eta }}_{y},{\tilde{\eta }}_{z}$ and ${\tilde{\eta }}_{\psi }$ are the results from the steady state error that asymptotically converge to values close to zero, i.e., since these position errors are bounded and different from zero $\eta_{\mathrm{ref}}\left(t\right)-W_{t}x\left(t\right)=0\in R^{3}$, it is achieved that the errors in the steady state are $|\tilde{\eta }|<0.15\left[m\right]$. In addition, the control error maintains a dependency on the external disturbances in the outdoor environments, and external disturbances are the product of the wind velocity which was approximately 10 km/h.

The time required to solve the optimal control problem is shown in Figure 14, and the computation time stays always below the sample time $t_{s}=0.1\;\left[s\right]$, guaranteeing the efficient computation of the proposed control considering the time horizon of T=2[s] in the optimal control problem.

### 3.2. Simulation Experiments

This section presents the simulation result of the proposed controller. In order to improve the results, this experiment was performed using the identified dynamics in Equation (Equation 36) with the identified parameters presented in Table 2 and Table 3. Furthermore, a Gaussian noise with the following consideration in all the states $\eta_{n}∼N\left(-0.05,0.05\right)$ and $\mu_{n}∼N\left(-0.01,0.01\right)$, where $\eta_{n}$ and $\mu_{n}$ are the additive noise in position and velocity states, was identified.

The desired reference trajectories and initial values over the frame <I> are defined in Table 6.

The desired values of the reference trajectory are the same as the proposed in real-world experiments; therefore, the results present similar behavior. The controller constant values are the same as those presented in Table 5. However, the safe region distance was defined as $r_{obs}=0.4\left[m\right]$.

The trajectory described by the hexacopter platform and the reference trajectory are presented in Figure 15. The results show that the hexacopter tracks the desired reference trajectory, in addition to the avoidance obstacles property being demonstrated due to the system following the reference and trying to maintain the distance to obstacles considering the safety radius $r_{obs}$. The three obstacles were positioned over the reference trajectory, two of which were positioned at the corners are dynamic obstacles and they have movements during the experiment.

The values of the control errors presented in Figure 16 tend to increase when the hexacopter avoids the obstacles satisfying the system constraints, a phenomenon which is represented in the rectangles representing obstacles close to the hexarotor. Given that the errors in steady state ${\tilde{\eta }}_{x},{\tilde{\eta }}_{y},{\tilde{\eta }}_{z}$ and ${\tilde{\eta }}_{\psi }$ symptomatically converge towards zero in the presence of the Gaussian noise applied in the simulation, the results show the robustness under this type of disturbance and the ability to keep the system under the reference trajectory, i.e., the position errors are bounded and are different from zero $\eta_{\mathrm{ref}}\left(t\right)-W_{t}x\left(t\right)=0$, achieving a final characteristic error with a max $|\tilde{\eta }|<0.9\left[m\right]$ when the obstacles are close to the hexarotor and $|\tilde{\eta }|<0.1\left[m\right]$ without obstacles near the system reference trajectory.

The control signals are presented in Figure 17, which are generated by the proposed control scheme. Taking into consideration the dynamics of the robotic system and the simulated environmental conditions, the behavior of the reference velocities are close to the dynamic model values. Furthermore, the values are presented as smooth curves with the presence of peaks when the obstacles are close to the hexacopter which are emphasized by the rectangles shown in Figure 17. The behavior of the control actions prevents collisions between the hexacopter and the obstacles during the simulation.

The time required to solve the proposed controller and the Euclidean distance are presented in Figure 18, and the computational time is an important factor to demonstrate its ability to generate an adequate control policy that guarantees the system constraints. The computational time presents peaks produced by the obstacles that are close to the hexacopter during the trajectory whilst the controller finds a sub-optimal solution to guarantee that the computational time remains under the sample time; however, this problem can be efficiently solved using another low-level language to implement the proposed controller or with a more powerful computer, which is therefore a solvable technological issue. On the other hand, the distance to each obstacle shows that the proposed controller respects the safely radius, guaranteeing the avoidance property of the system.

### 3.3. Discussion

The experiments presented in the last section show that the nonlinear model predictive scheme solves the control problem associated with the trajectory tracking subject to the system and environment constraints. Specifically, the real-world experiments showed the robustness of the proposed controller due to the excessive wind velocity during the experiment which acted as an external disturbance. Despite the excessive wind velocity, the controller generates the adequate smooth control actions guaranteeing that the steady state error that converges towards values close to zero with the dependency of the external disturbances. The use of CasADI such as an optimization framework guarantees the fact that the computational time always stays below the sample time, which is one of main problems in NMPC schemes due to the highly computational time that these solutions required, as similar computing times were presented in [45].

On the other hand, the simulation results show the ability of the proposed scheme to generates the adequate control signals guaranteeing the corrected tracking trajectories while the system avoids obstacles during the simulation. The presence of Gaussian noise provides uncertainty in the measurements demonstrating the power of the controller as it maintains the distance to obstacles in consideration of the safety radius. One of the main differences presented in this experiment was the control error, which tends to increase when the hexacopter avoids obstacles satisfying the system constraints. Furthermore, the computational time presents peaks produced by the hard constraints included in the optimization problem, and despite the hard constraint, the controller finds a sub-optimal solution to guarantee a low computational time. Other works have used the concept of soft constraints, low-level programming language and embeddable optimization methods such as (SQP) in order to increase the convergence of the optimization algorithm and reduce the computational time [60,61,62,63,64]. This work uses the concept of soft constraints and the approximation of the nonlinear problem in order to improve the computational time and the future implementation on a single on-board PC.

## 4. Conclusions

This work carried out the identification of a dynamic system using the dynamic mode decomposition with control technique to develop the optimal control problem for trajectory tracking with obstacle avoidance, a decision which was made due to the precision of the comparative results between the Euler–Lagrange and the dynamic mode decomposition formulations. The optimal control problem was translated into a nonlinear programming problem (NLP) using the multiple shooting technique and considering the control actions and systems states as decision variables, which is a technique that improves the computational time and the convergence of the algorithm. The nonlinear model predictive control (NMPC) generates the maneuverability velocities policy that allows the hexacopter platform to track the reference trajectory while avoiding obstacles presented in the environment. The NMPC formulation was developed in consideration of the reference desired trajectory and constraints in the system such as the bounded control actions, the generalized dynamics and static and dynamic obstacles. Furthermore, the NMPC problem was solved through the nonlinear programming framework CasADI, which solves high-dimensional optimization problems and has advantages that can be expanded to a low-level programming language to improve the computational time.

The simulation results show how the hexacopter avoids both static and dynamic obstacles whose locations vary over time while executing the trajectory tracking task, and always maintaining a safe distance defined as a radius of repulsion. Since there is no sensor for obstacle measurements in the hexacopter platform, the real experimental tests were not carried out with obstacle avoidance; however it was verified that although the experimental tests were carried out during the period of time in which the average wind speed was approximately 10 km/h, the behavior of the control law on the hexacopter system fulfills the desired task, so that the control errors in stable state converge closer to zero. In addition, it can be observed that the steady state error asymptotically converges towards values closer to zero under conditions wherein the control actions are saturate when they reach the established restrictions. On the other hand, the computational time of the proposed control algorithm is maintained under 100 ms in real experimentation due to the optimization technique used to solve the control problem. During the simulation tests, it is clearly verified that the computational time remains under 100 ms; however, when obstacle avoidance occurs, this time increases but remains under the specified sample time with a sup-optimal solution. Other works use the concept of soft constraints to guarantee the rapid convergence of the optimization algorithm, as this work will use the same concept to improve the computational time and the possible expansion to visual-servoing systems under an MPC structure.

## Figures and Tables

**Figure 1 sensors-22-04712-f001:**
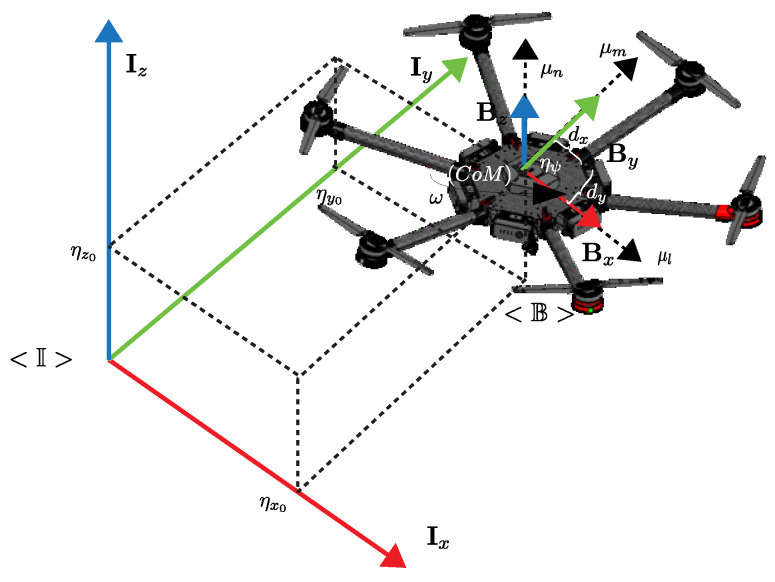
Hexacopter platform representation.

**Figure 2 sensors-22-04712-f002:**
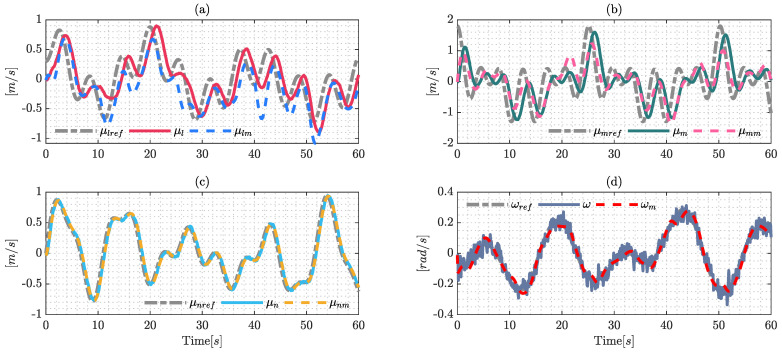
Signals of the identification process using the Euler-Lagrange formulation with the optimization techniques (SQPs). The results are confirmed by: (**a**) identification of the reference velocity $\mu_{lref}$; (**b**) shows the results over the reference velocity $\mu_{mref}$; (**c**) identification over the upper longitudinal velocity $\mu_{nref}$; and (**d**) shows the results of the reference angular velocity $\omega_{ref}$.

**Figure 3 sensors-22-04712-f003:**
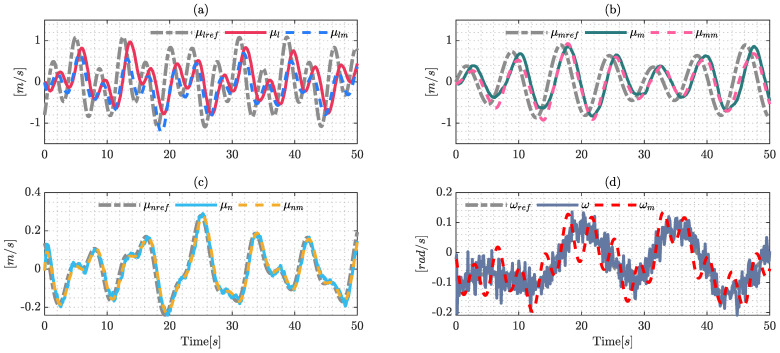
Validation process signals using the Euler–Lagrange formulation with the optimization technique (SQP). The results are confirmed by: (**a**) validation over the reference velocity $\mu_{lref}$; (**b**) shows the results over the reference velocity $\mu_{mref}$; (**c**) shows the validation over the longitudinal velocity $\mu_{nref}$; and (**d**) shows the validation results of the reference angular velocity $\omega_{ref}$.

**Figure 4 sensors-22-04712-f004:**
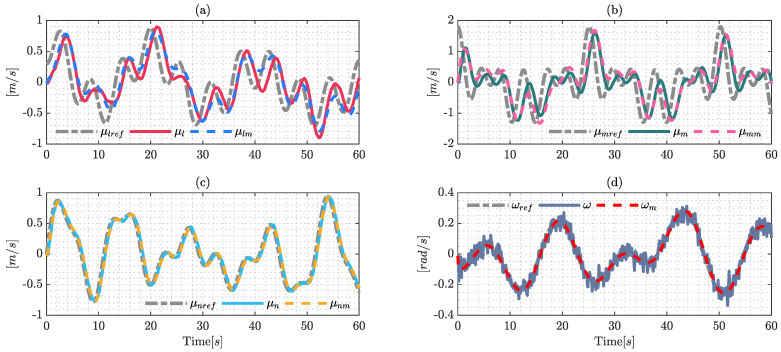
Identification process using dynamic mode decomposition formulation (DMD): (**a**) results over $\mu_{lref}$; (**b**) identification on $\mu_{mref}$; (**c**) validation over $\mu_{nref}$; and (**d**) shows the results on $\omega_{ref}$.

**Figure 5 sensors-22-04712-f005:**
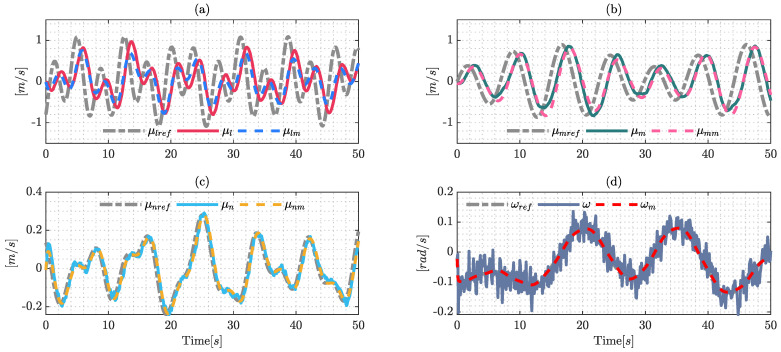
Validation process using dynamic mode decomposition (DMD). The results are: (**a**) validation over the reference velocity $\mu_{lref}$; (**b**) shows the results through velocity $\mu_{mref}$; (**c**) validation over the longitudinal velocity $\mu_{nref}$; and (**d**) shows the validation results of the reference angular velocity $\omega_{ref}$.

**Figure 6 sensors-22-04712-f006:**
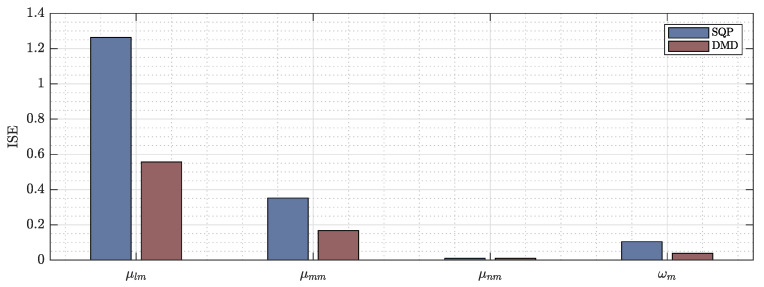
Integral square error of the identified models using Euler–Lagrange with sequential quadratic programming (SQP) and dynamic mode decomposition (DMD) formulation.

**Figure 7 sensors-22-04712-f007:**
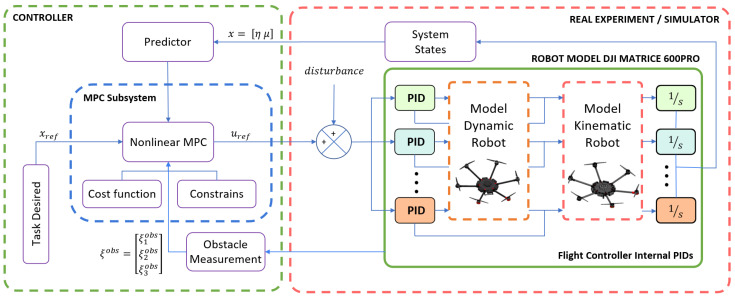
Proposed controller scheme.

**Figure 8 sensors-22-04712-f008:**
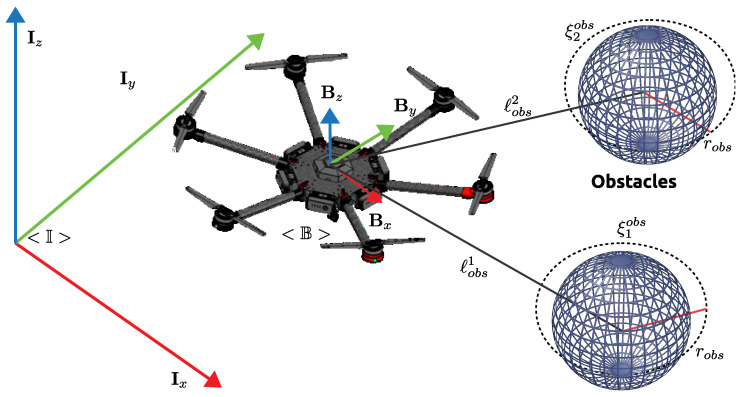
Representation of the hexacopter and obstacles.

**Figure 9 sensors-22-04712-f009:**
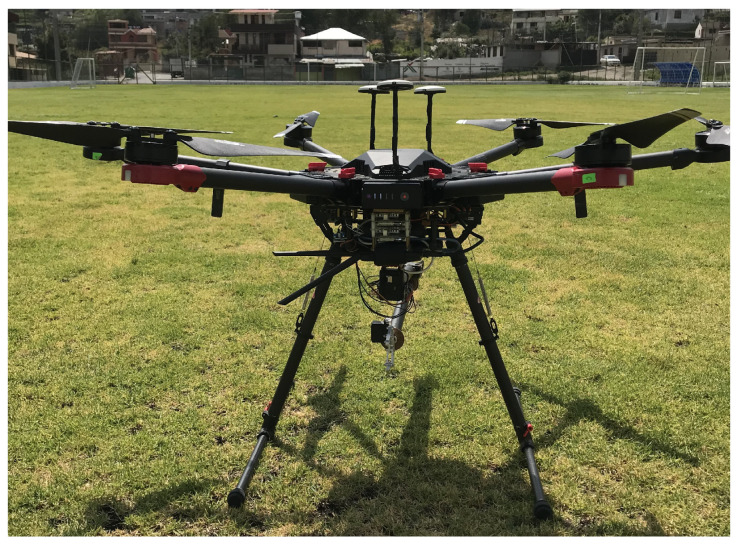
Hexacopter platform used in a real-world experiment.

**Figure 10 sensors-22-04712-f010:**
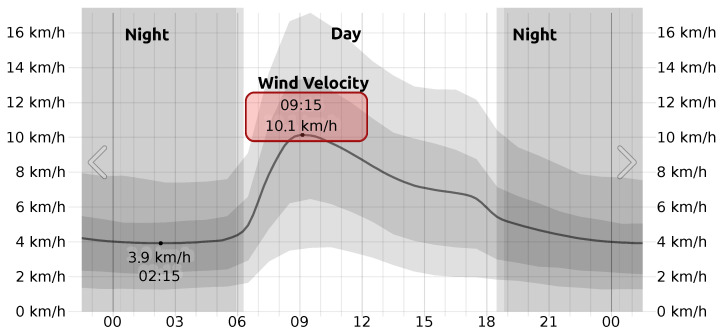
Wind velocity during experiment.

**Figure 11 sensors-22-04712-f011:**
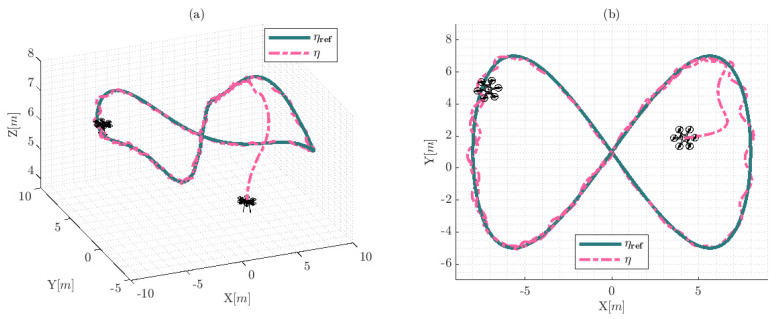
Movements of the hexacopter based on the real-world experimentation: (**a**) represents the behavior of the hexacopter platform using the 3D isometric perspective $\left\{I_{x},I_{y},I_{z}\right\}$; and (**b**) shows the upper view $\left\{I_{x},I_{y}\right\}$ of the experimental information.

**Figure 12 sensors-22-04712-f012:**
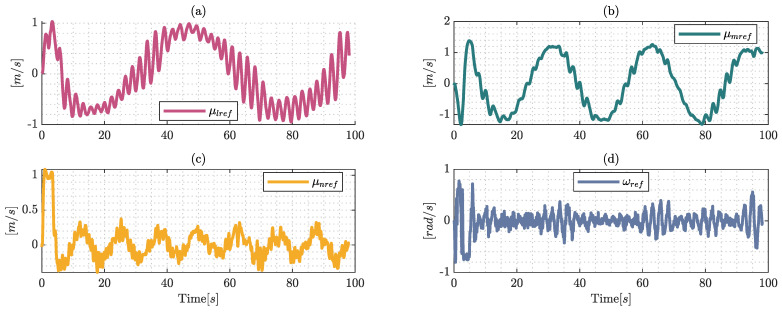
Control signals generated by the proposed control during the real-world experiment, where (**a**) represents the control velocity over the axis $\left\{I_{x}\right\}$; (**b**) describes the evolution of velocity in $\left\{I_{y}\right\}$; finally (**c**) and (**d**) represent the evolution of the upper and angular control velocity, respectively.

**Figure 13 sensors-22-04712-f013:**
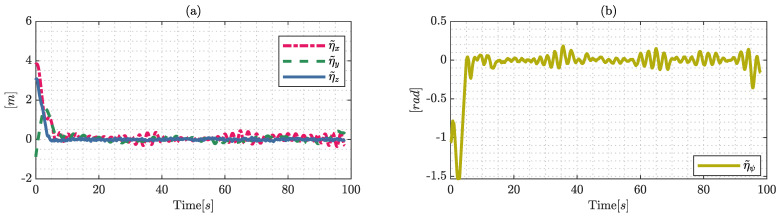
Control errors during a real outdoor experiment: (**a**) represents the error in the work-space $\left\{I_{x},I_{y},I_{z}\right\}$; and (**b**) shows the results with respect to the desired angular position.

**Figure 14 sensors-22-04712-f014:**
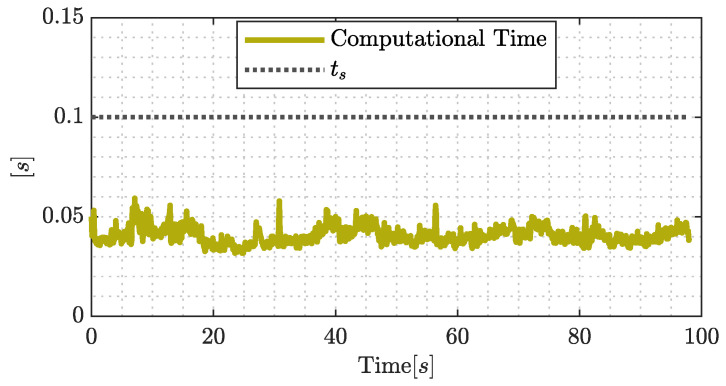
Computational time during the real-world experimental results.

**Figure 15 sensors-22-04712-f015:**
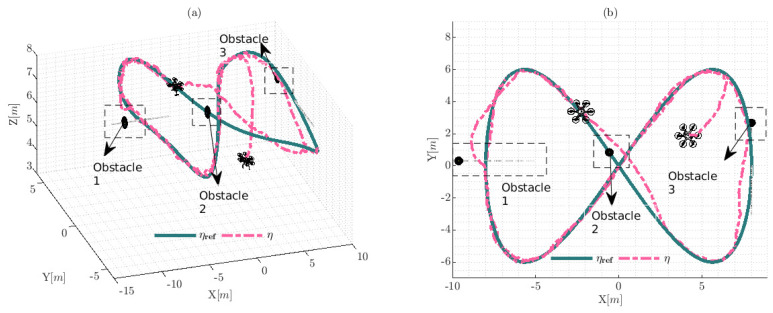
Movements of the hexacopter during the simulation experimentation: (**a**) represents the behavior of the hexacopter platform using the 3D isometric perspective $\left\{I_{x},I_{y},I_{z}\right\}$; and (**b**) shows the upper view of the experimental information; furthermore, the black spheres represent the static and dynamic obstacles. The velocity of the dynamic obstacles was approximately 0.2cos(0.1t)[m/s] over $I_{x}$ for the first obstacle and approximately 0.6cos(0.2t)[m/s] over $I_{y}$ for the third obstacle. The second obstacle was considered static and the position is static during the experiment.

**Figure 16 sensors-22-04712-f016:**
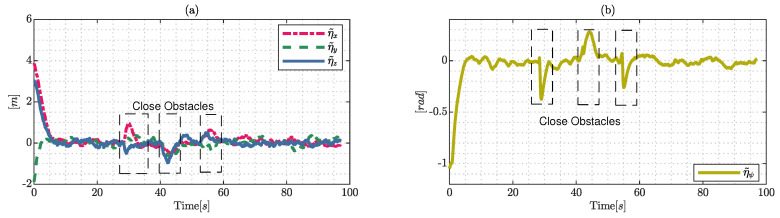
Control errors during the simulated experiment: (**a**) represents the error in the work-space; and (**b**) shows the error with respect to angular displacements.

**Figure 17 sensors-22-04712-f017:**
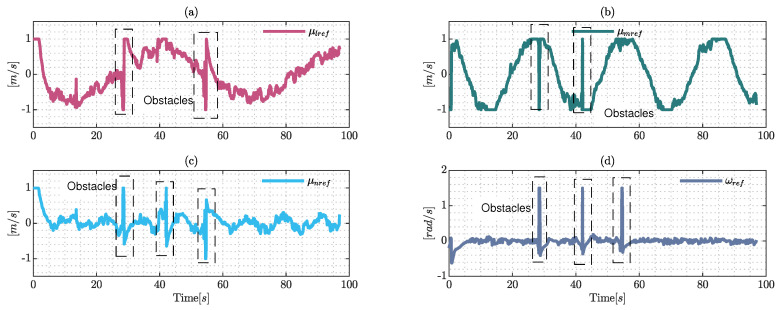
Control signals generated by the proposed control and maneuverability velocities were generated by the dynamic approximation, where (**a**) represents the maneuverability velocities over the axis $\left\{I_{x}\right\}$; (**b**) describes the evolution of velocities in $\left\{I_{y}\right\}$; finally, (**c**) and (**d**) represent the evolution of the upper and angular velocity, respectively.

**Figure 18 sensors-22-04712-f018:**
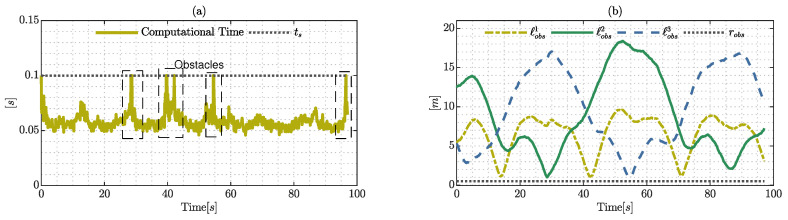
Computational time and distance to obstacles: (**a**) represents the time to solve the optimization problem; and (**b**) shows the distance to the obstacles, where $\ell_{obs}^{1}$$\ell_{obs}^{2}$ and $\ell_{obs}^{3}$ are the respective distances and $r_{obs}$ is the radius of security considering the dimensions of the hexarotor frame.

**Table 1 sensors-22-04712-t001:** Dynamic parameters of the hexacopter aerial platform.

System	Dynamic Parameters		
DJI MATRICE 600 PRO	$\zeta_{1}=2.11$	$\zeta_{2}=-0.005$	$\zeta_{3}=1.8$
	$\zeta_{4}=3.17$	$\zeta_{5}=1.78$	$\zeta_{6}=0.39$
	$\zeta_{7}=-0.003$	$\zeta_{8}=-0.03$	$\zeta_{9}=0.006$
	$\zeta_{10}=0.02$	$\zeta_{11}=0.002$	$\zeta_{12}=0.06$
	$\zeta_{13}=0.70$	$\zeta_{14}=0.02$	$\zeta_{15}=-0.05$
	$\zeta_{16}=-0.01$	$\zeta_{17}=-0.005$	$\zeta_{18}=-0.01$
	$\zeta_{19}=0.831$		

**Table 2 sensors-22-04712-t002:** Approximation values of unforced matrix.

Matrix	Approximated Values			
$\bar{A}$	$a_{11}=-0.5827$	$a_{12}=-0.0721$	$a_{13}=0.0587$	$a_{14}=-0.0280$
	$a_{21}=0.0214$	$a_{22}=-0.4770$	$a_{23}=0.0731$	$a_{24}=-0.1840$
	$a_{31}=0.0152$	$a_{32}=0.0102$	$a_{33}=-3.5696$	$a_{34}=0.0064$
	$a_{41}=0.0377$	$a_{42}=-0.0113$	$a_{43}=-0.0058$	$a_{44}=-8.2424$

**Table 3 sensors-22-04712-t003:** Approximate values of control actuation matrix.

Matrix	Approximated Values			
$\bar{B}$	$b_{11}=0.8048$	$b_{12}=-0.0376$	$b_{13}=-0.0793$	$b_{14}=-0.1797$
	$b_{21}=0.0634$	$b_{22}=0.8527$	$b_{23}=-0.0783$	$b_{24}=0.3821$
	$b_{31}=-0.0202$	$b_{32}=-0.0046$	$b_{33}=3.5776$	$b_{34}=0.0607$
	$b_{41}=-0.0251$	$b_{42}=-0.0029$	$b_{43}=-0.0288$	$b_{44}=8.1294$

**Table 4 sensors-22-04712-t004:** Desired reference trajectory in a real-world experiment.

Initial Positions	Initial Maneuverability Velocities	Reference Trajectory
$\eta_{x_{o}}=4.18\left[m\right]$	$\mu_{l_{o}}=0.01\left[m/s\right]$	$\eta_{x_{ref}}=8\sin\left(0.1t\right)\left[m\right]$
$\eta_{y_{o}}=1.88\left[m\right]$	$\mu_{m_{o}}=-0.04\left[m/s\right]$	$\eta_{y_{ref}}=6\sin\left(0.2t\right)+1\left[m\right]$
$\eta_{z_{o}}=3.97\left[m\right]$	$\mu_{n_{o}}=-0.02\left[m/s\right]$	$\eta_{z_{ref}}=0.35\sin\left(0.5t\right)+7\left[m\right]$
$\eta_{\psi_{o}}=1.05\left[\mathrm{rad}\right]$	$\omega_{o}=-0.02\left[\mathrm{rad}/s\right]$	$\eta_{\psi_{ref}}=0\left[\mathrm{rad}\right]$

**Table 5 sensors-22-04712-t005:** Proposed values for the NMPC scheme in a real-world experiment.

Parameters	Values	Parameters	Values
$Q_{T}$	$diag\left(1\right)\in R^{4\times 4}$	$Q_{t}$	$diag\left(1\right)\in R^{4\times 4}$
$Q_{u}$	$diag\left(0.005\right)\in R^{4\times 4}$	*T*	2[s]
$\mu_{{\mathrm{ref}}_{min}}$	−[1,1,1,1.5][m/s,rad/s]	$\mu_{{\mathrm{ref}}_{max}}$	[1,1,1,1.5][m/s,rad/s]
$\Delta \mu_{{\mathrm{ref}}_{max}}$	0.05[m/s,rad/s]		

**Table 6 sensors-22-04712-t006:** Desired reference trajectory in the simulation experiment.

Initial Positions	Initial Maneuverability Velocities	Reference Trajectory
$\eta_{x_{o}}=4.18\left[m\right]$	$\mu_{l_{o}}=0\left[m/s\right]$	$\eta_{x_{ref}}=8\sin\left(0.1t\right)\left[m\right]$
$\eta_{y_{o}}=1.88\left[m\right]$	$\mu_{m_{o}}=0\left[m/s\right]$	$\eta_{y_{ref}}=6\sin\left(0.2t\right)+1\left[m\right]$
$\eta_{z_{o}}=3.97\left[m\right]$	$\mu_{n_{o}}=0\left[m/s\right]$	$\eta_{z_{ref}}=0.35\sin\left(0.5t\right)+7\left[m\right]$
$\eta_{\psi_{o}}=1.05\left[\mathrm{rad}\right]$	$\omega_{o}=0\left[\mathrm{rad}/s\right]$	$\eta_{\psi_{ref}}=0\left[\mathrm{rad}\right]$

## Data Availability

If anyone wants to obtain the original data of this paper, please contact with Luis F. Recalde.

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
