# Peer review of "System Identification and Nonlinear Model Predictive Control with Collision Avoidance Applied in Hexacopters UAVs"

_sensors, 2022, doi:10.3390/s22134712_

Round 1
Reviewer 1 Report
The manuscript is dedicated to nonlinear model predictive control with collision avoidance for tracking the hexacopters trajectory with obstacle avoidance in dynamic environments. The experimental results prove the good performance of the predictive scheme and its ability to regenerate the optimal control policy. The paper is written well, but it should be reorganized to be in accordance with the journal requirements to the structure of the article. The authors have provided a detailed Introduction, describing a deep overview of the subject, and indicating the need for their study. The results obtained are new, interesting, valuable, and presented clearly, but they need to be discussed in section Discussion. Unfortunately, this section is missing in the manuscript. My opinion is that the paper needs at least minor revision. The suggested corrections are below.
Corrections suggested.
- The paper should have the following structure (see the journal template): Introduction, Materials and Methods, Results, Discussion, and Conclusions. Sections Materials and Methods, Results, Discussion are missing. Please, reorganize your paper in accordance with the required structure.
- The article contains about 50 equations. If some of them are from other articles or other sources, please, provide appropriate references.
- Page 10. Line 290.
Please, write “Appendix A” instead of “Appendix”.
- References.
Please, prepare all the references in accordance with the journal requirements exactly within the journal template and abbreviate properly titles of journals.
Please, provide correct DOI for Reference [2] instead of “https://doi.org/10.1007/978-981-10-6451-7{\_}33”.
Please, provide correct DOI for Reference [12] instead of “https://doi.org/10.1007/978-3-642-32723-0{\_}39”.
- Please, rewrite Conclusions providing clearer information on the results obtained. Also, describe in Conclusion where the results obtained can be used.
Author Response
Dear reviewer, the authors appreciate the corrections made. Below are the responses to your comments.
The paper should have the following structure (see the journal template): Introduction, Materials and Methods, Results, Discussion, and Conclusions. Sections Materials and Methods, Results, Discussion are missing. Please, reorganize your paper in accordance with the required structure.
Response:
It’s done, the structure of the paper was remade
The article contains about 50 equations. If some of them are from other articles or other sources, please, provide appropriate references.
Response:
It’s done. All equations were developed in this work, however just the basic ideas were obtained from previous works.
Page 10. Line 290.
Please, write “Appendix A” instead of “Appendix”.
Response:
It’s done, p10/r290
Please, prepare all the references in accordance with the journal requirements exactly within the journal template and abbreviate properly titles of journals.
Response:
It’s done, we verified each reference and the problem with references [2] and [12] were solved.
Please, rewrite Conclusions providing clearer information on the results obtained. Also, describe in Conclusion where the results obtained can be used.
Response:
It’s done, the conclusion has been corrected based on this recommendation.
Reviewer 2 Report
The paper describes the application of nonlinear model predictive control with collision avoidance in hexacopter real model and also experimental identification of its dynamic parameters based on an accuracy comparative study. The paper is very well written in a correct and easily understandable way. The literature review is perfect and allows an outlook on other solutions. Authors described very well state of art and mathematical and theoretical background of investigated problem. I appreciate the detailed analysis of the problems and the reasons for solving problems related to the investigated issues in this paper. The achieved results have been experimentally validated but in my opinion the novelty of the proposed solutions is not highlighted enough in the paper.
In conclusion the proposed method should be more deeply compared with other ones cited in the literature, and its advantages should be emphasized.
One of the paper´s drawback is that it lacks description of the hardware hexacopter platform, for example its sensorial system. It is important to describe how the experimental measurements for system identification have been obtained.
To eliminate noise in real measurement a filter described in page11/row 304 has been used, where value of parameter “lambda” is missing.
The end of the abstract should be rewritten (rows 13-16). The same idea seems to be described here.
For better clarity and comprehensibility, it is necessary to unify the writing of variables in equations, text and figures. This means using the same font to denote variables in equations, formulas and figures.
Also English should be corrected on some places (p4/r177 “… to shows…”, p13/r312 “…platforms cannot be controler? …”, p17/r382 “… the euclidean norm,…!, p26/r536 “… and show how? …”)
Author Response
Dear reviewer, the authors appreciate the corrections made. Below are the responses to your comments.
In conclusion the proposed method should be more deeply compared with other ones cited in the literature, and its advantages should be emphasized.
Response:
The conclusion has been corrected based on this recommendation.
One of the paper´s drawback is that it lacks description of the hardware hexacopter platform, for example its sensorial system. It is important to describe how the experimental measurements for system identification have been obtained.
Response:
It’s done, p11/r297
To eliminate noise in real measurement a filter described in page11/row 304 has been used, where value of parameter “lambda” is missing.
Response:
It’s done, p11/r307
The end of the abstract should be rewritten (rows 13-16). The same idea seems to be described here.
Response:
It’s done, p1/r13
For better clarity and comprehensibility, it is necessary to unify the writing of variables in equations, text and figures. This means using the same font to denote variables in equations, formulas and figures.
Response:
My apologies but we can't change the font in the figures, the software that we used does not have this option
Also English should be corrected on some places (p4/r177 “… to shows…”, p13/r312 “…platforms cannot be controller? …”, p17/r382 “… the euclidean norm,…!, p26/r536 “… and show how? …”)
Response:
It’s done
Reviewer 3 Report
This paper presents a nonlinear model and predictive control for a hexacopter, capable of tracking a planned route while avoiding obstacles it may encounter along the way.
In general, the paper is well structured. The introduction and the description of the state of the art are correct. The methodology used is rigorous and the conclusions are supported by the results obtained. English is also correct.
Regarding the technical aspect, the paper describes the proposed kinematic and dynamic models for the drone, and their subsequent instantiation from empirical data obtained from a real aircraft.
Subsequently, trajectory control is addressed. Being developed in the CasADI optimization framework, this section is limited to formulating the cost function with the corresponding maneuverability constraints of the device, to which constraints generated by the presence of obstacles are added. Let me suggest the inclusion in section 3.1.4 of the nonlinear optimal control provided by this tool, since, after all the detail invested in the previous formulation, the reader does not get to see the final result.
Author Response
Dear reviewer, the authors appreciate the corrections made. Below are the responses to your comments.
Let me suggest the inclusion in section 3.1.4 of the nonlinear optimal control provided by this tool, since, after all the detail invested in the previous formulation, the reader does not get to see the final result.
Response:
The optimal control part has been corrected based on this recommendation